# Polynomial Width is Sufficient for Set Representation with High-dimensional Features

## Abstract

Set representation has become ubiquitous in deep learning for modeling the inductive bias of neural networks that are insensitive to the input order. DeepSets is the most widely used neural network architecture for set representation. It involves embedding each set element into a latent space with dimension $L$, followed by a sum pooling to obtain a whole-set embedding, and finally mapping the whole-set embedding to the output. In this work, we investigate the impact of the dimension $L$ on the expressive power of DeepSets. Previous analyses either oversimplified high-dimensional features to be one-dimensional features or were limited to analytic activations, thereby diverging from practical use or resulting in $L$ that grows exponentially with the set size $N$ and feature dimension $D$. To investigate the minimal value of $L$ that achieves sufficient expressive power, we present two set-element embedding layers: (a) linear + power activation (LP) and (b) logarithm + linear + exponential activations (LLE). We demonstrate that $L$ being $\mathrm{poly}(N, D)$ is sufficient for set representation using both embedding layers. We also provide a lower bound of $L$ for the LP embedding layer. Furthermore, we extend our results to permutation-equivariant set functions and the complex field.

## 1  Introduction

Enforcing invariance into neural network architectures has become a widely-used principle to design deep learning models [1–7]. In particular, when a task is to learn a function with a set as the input, the architecture enforces permutation invariance that asks the output to be invariant to the permutation of the input set elements [8, 9]. Neural networks to learn a set function have found a variety of applications in particle physics [10, 11], computer vision [12, 13] and population statistics [14–16], and have recently become a fundamental module (the aggregation operation of neighbors' features in a graph [17–19]) in graph neural networks (GNNs) [20, 21] that show even broader applications.

Previous works have studied the expressive power of neural network architectures to represent set functions [8, 9, 22–26]. Formally, a set with $N$ elements can be represented as $\mathcal{S} = \{\boldsymbol{x}^{(1)}, \cdots, \boldsymbol{x}^{(N)}\}$ where $\boldsymbol{x}^{(i)}$ is in a feature space $\mathcal{X}$, typically $\mathcal{X} = \mathbb{R}^D$. To represent a set function that takes $\mathcal{S}$ and outputs a real value, the most widely used architecture DeepSets [9] follows Eq. (1).

$$f(\mathcal{S}) = \rho\left(\sum_{i=1}^{N} \phi(\boldsymbol{x}^{(i)})\right), \text{where } \phi : \mathcal{X} \to \mathbb{R}^L \text{ and } \rho : \mathbb{R}^L \to \mathbb{R} \text{ are continuous functions.} \quad (1)$$

DeepSets encodes each set element individually via $\phi$, and then maps the encoded vectors after sum pooling to the output via $\rho$. The continuity of $\phi$ and $\rho$ ensure that they can be well approximated by fully-connected neural networks [27, 28], which has practical implications. DeepSets enforces permutation invariance because of the sum pooling, as shuffling the order of $\boldsymbol{x}^{(i)}$ does not change

Submitted to 37th Conference on Neural Information Processing Systems (NeurIPS 2023). Do not distribute.

Table 1: A comprehensive comparison among all prior works on expressiveness analysis with $L$. Our results achieve the tightest bound on $L$ while being able to analyze high-dimensional set features and extend to the equivariance case.

| Prior Arts | $L$ | $D > 1$ | Exact Rep. | Equivariance |
|---|---|---|---|---|
| DeepSets [9] | $D + 1$ | ✗ | ✓ | ✓ |
| Wagstaff et al. [23] | $D$ | ✗ | ✓ | ✓ |
| Segol et al. [25] | $\binom{N+D}{N} - 1$ | ✓ | ✗ | ✓ |
| Zweig & Bruna [26] | $\exp(\min\{\sqrt{N}, D\})$ | ✓ | ✗ | ✗ |
| Our results | $\mathrm{poly}(N, D)$ | ✓ | ✓ | ✓ |

33  the output. However, the sum pooling compresses the whole set into an $L$-dimension vector, which
34  places an information bottleneck in the middle of the architecture. Therefore, a core question on
35  using DeepSets for set function representation is that given the input feature dimension $D$ and the
36  set size $N$, what the minimal $L$ is needed so that the architecture Eq. (1) can represent/universally
37  approximate any continuous set functions. The question has attracted attention in many previous
38  works [9, 23–26] and is the focus of the present work.

39  An extensive understanding has been achieved for the case with one-dimensional features ($D = 1$).
40  Zaheer et al. [9] proved that this architecture with bottleneck dimension $L = N$ suffices to *accurately*
41  represent any continuous set functions when $D = 1$. Later, Wagstaff et al. proved that accurate
42  representations cannot be achieved when $L < N$ [23] and further strengthened the statement to *a*
43  *failure in approximation* to arbitrary precision in the infinity norm when $L < N$ [24].

44  However, for the case with high-dimensional features ($D > 1$), the characterization of the minimal
45  possible $L$ is still missing. Most of previous works [9, 25, 29] proposed to generate multi-symmetric
46  polynomials to approximate permutation invariant functions [30]. As the algebraic basis of multi-
47  symmetric polynomials is of size $L^* = \binom{N+D}{N} - 1$ [31] (exponential in $\min\{D, N\}$), these works by
48  default claim that if $L \geq L^*$, $f$ in Eq. 1 can approximate any continuous set functions, while they do
49  not check the possibility of using a smaller $L$. Zweig and Bruna [26] constructed a set function that $f$
50  requires bottleneck dimension $L > N^{-2} \exp(O(\min\{D, \sqrt{N}\}))$ (still exponential in $\min\{D, \sqrt{N}\}$)
51  to approximate while it relies on the condition that $\phi, \rho$ only adopt analytic activations. This condition
52  is overly strict, as most of the practical neural networks allow using non-analytic activations, such as
53  ReLU. Zweig and Bruna thus left an open question *whether the exponential dependence on $N$ or $D$*
54  *of $L$ is still necessary if $\phi, \rho$ allow using non-analytic activations.*

55  **Present work** The main contribution of this work is to confirm a negative response to the above
56  question. Specfically, we present the first theoretical justification that $L$ being *polynomial* in $N$ and
57  $D$ is sufficient for DeepSets (Eq. (1)) like architecture to represent any *continuous* set functions
58  with *high-dimensional* features ($D > 1$). To mitigate the gap to the practical use, we consider two
59  architectures to implement feature embedding $\phi$ (in Eq. 1) and specify the bounds on $L$ accordingly:

60  • $\phi$ adopts *a linear layer with power mapping*: The minimal $L$ holds a lower bound and an upper
61  bound, which is $N(D + 1) \leq L < N^5 D^2$.

62  • Constrained on the entry-wise positive input space $\mathbb{R}_{>0}^{N \times D}$, $\phi$ adopts *two layers with logarithmic*
63  *and exponential activations respectively*: The minimal $L$ holds a tighter upper bound $L \leq 2N^2 D^2$.

64  We prove that if the function $\rho$ could be any continuous function, the above two architectures
65  reproduce the precise construction of any set functions for high-dimensional features $D > 1$, akin
66  to the result in [9] for $D = 1$. This result contrasts with [25, 26] which only present approximating
67  representations. If $\rho$ adopts a fully-connected neural network that allows approximation of any
68  continuous functions on a bounded input space [27, 28], then the DeepSets architecture $f(\cdot)$ can
69  approximate any set functions universally on that bounded input space. Moreover, our theory can be
70  easily extended to permutation-equivariant functions and complex set functions, where the minimal
71  $L$ shares the same bounds up to some multiplicative constants.

72  Another comment on our contributions is that Zweig and Bruna [26] use difference in the needed
73  dimension $L$ to illustrate the gap between DeepSets [9] and Relational Network [32] in their expressive
74  powers, where the latter encodes set elements in a pairwise manner rather than in a separate manner.
75  The gap well explains the empirical observation that Relational Network achieves better expressive
76  power with smaller $L$ [23, 33]. Our theory does not violate such an observation while it shows that the

gap can be reduced from an exponential order in $N$ and $D$ to a polynomial order. Moreover, many real-world applications have computation constraints where only DeepSets instead of Relational Network can be used, e.g., the neighbor aggregation operation in GNN being applied to large networks [21], and the hypergraph neural diffusion operation in hypergraph neural networks [7]. Our theory points out that in this case, it is sufficient to use polynomial $L$ dimension to embed each element, while one needs to adopt a function $\rho$ with non-analytic activitions.

## 2 Preliminaries

### 2.1 Notations and Problem Setup

We are interested in the approximation and representation of functions defined over sets [1]. In convention, an $N$-sized set $\mathcal{S} = \{\boldsymbol{x}^{(1)}, \cdots, \boldsymbol{x}^{(N)}\}$, where $\boldsymbol{x}^{(i)} \in \mathbb{R}^D, \forall i \in [N] (\triangleq \{1, 2, ..., N\})$, can be denoted by a data matrix $\boldsymbol{X} = \begin{bmatrix} \boldsymbol{x}^{(1)} & \cdots & \boldsymbol{x}^{(N)} \end{bmatrix}^\top \in \mathbb{R}^{N \times D}$. Note that we use the superscript $(i)$ to denote the $i$-th set element and the subscript $i$ to denote the $i$-th column/feature channel of $\boldsymbol{X}$, i.e., $\boldsymbol{x}_i = \begin{bmatrix} x_i^{(1)} & \cdots & x_i^{(N)} \end{bmatrix}^\top$. Let $\Pi(N)$ denote the set of all $N$-by-$N$ permutation matrices. To characterize the unorderedness of a set, we define an equivalence class over $\mathbb{R}^{N \times D}$:

**Definition 2.1** (Equivalence Class). If matrices $\boldsymbol{X}, \boldsymbol{X'} \in \mathbb{R}^{N \times D}$ represent the same set $\mathcal{X}$, then they are called equivalent up a row permutation, denoted as $\boldsymbol{X} \sim \boldsymbol{X'}$. Or equivalently, $\boldsymbol{X} \sim \boldsymbol{X'}$ if and only if there exists a matrix $\boldsymbol{P} \in \Pi(N)$ such that $\boldsymbol{X} = \boldsymbol{P}\boldsymbol{X'}$.

Set functions can be in general considered as permutation-invariant or permutation-equivariant functions, which process the input matrices regardless of the order by which rows are organized. The formal definitions of permutation-invariant/equivariant functions are presented as below:

**Definition 2.2.** (Permutation Invariance) A function $f : \mathbb{R}^{N \times D} \to \mathbb{R}^{D'}$ is called permutation-invariant if $f(\boldsymbol{P}\boldsymbol{X}) = f(\boldsymbol{X})$ for any $\boldsymbol{P} \in \Pi(N)$.

**Definition 2.3.** (Permutation Equivariance) A function $f : \mathbb{R}^{N \times D} \to \mathbb{R}^{N \times D'}$ is called permutation-equivariant if $f(\boldsymbol{P}\boldsymbol{X}) = \boldsymbol{P}f(\boldsymbol{X})$ for any $\boldsymbol{P} \in \Pi(N)$.

In this paper, we investigate the approach to design a neural network architecture with permutation invariance/equivariance. Below we will first focus on permutation-invariant functions $f : \mathbb{R}^{N \times D} \to \mathbb{R}$. Then, in Sec. 5, we show that we can easily extend the established results to permutation-equivariant functions through the results provided in [7, 34] and to the complex field. The obtained results for $D' = 1$ can also be easily extended to $D' > 1$ as otherwise $f$ can be written as $\begin{bmatrix} f_1 & \cdots & f_{D'} \end{bmatrix}^\top$ and each $f_i$ has single output feature channel.

### 2.2 DeepSets and The Difficulty in the High-Dimensional Case $D > 1$

The seminal work [9] establishes the following result which induces a neural network architecture for permutation-invariant functions.

**Theorem 2.4** (DeepSets [9], $D = 1$). *A continuous function $f : \mathbb{R}^N \to \mathbb{R}$ is permutation-invariant (i.e., a set function) if and only if there exists continuous functions $\phi : \mathbb{R} \to \mathbb{R}^L$ and $\rho : \mathbb{R}^L \to \mathbb{R}$ such that $f(\boldsymbol{X}) = \rho\left(\sum_{i=1}^N \phi(x^{(i)})\right)$, where $L$ can be as small as $N$. Note that, here $x^{(i)} \in \mathbb{R}$.*

*Remark* 2.5. The original result presented in [9] states the latent dimension should be as large as $N + 1$. [23] tighten this dimension to exactly $N$.

Theorem 2.4 implies that as long as the latent space dimension $L \geq N$, any permutation-invariant functions can be implemented by a unified manner as DeepSets (Eq.(1)). Furthermore, DeepSets suggests a useful architecture for $\phi$ at the analysis convenience and empirical utility, which is formally defined below ($\phi = \psi_L$):

**Definition 2.6** (Power mapping). A power mapping of degree $K$ is a function $\psi_K : \mathbb{R} \to \mathbb{R}^K$ which transforms a scalar to a power series: $\psi_K(z) = \begin{bmatrix} z & z^2 & \cdots & z^K \end{bmatrix}^\top$.

---

[1] In fact, we allow repeating elements in $\mathcal{S}$, therefore, $\mathcal{S}$ should be more precisely called multiset. With a slight abuse of terminology, we interchangeably use terms multiset and set throughout the whole paper.

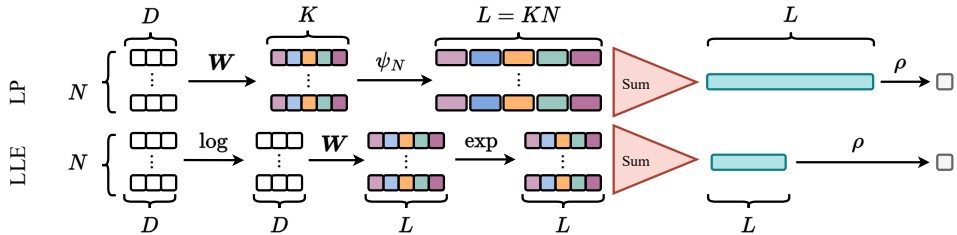

Figure 1: Illustration of the proposed linear + power mapping embedding layer (LP) and logarithm activation + linear + exponential activation embedding layer (LLE).

However, DeepSets [9] focuses on the case that the feature dimension of each set element is one (i.e., $D = 1$). To demonstrate the difficulty extending Theorem 2.4 to high-dimensional features, we reproduce the proof next, which simultaneously reveals its significance and limitation. Some intermediate results and mathematical tools will be recalled along the way later in our proof.

We begin by defining sum-of-power mapping (of degree $K$) $\Psi_K(\boldsymbol{X}) = \sum_{i=1}^{N} \psi_K(x_i)$, where $\psi_K$ is the power mapping following Definition 2.6. Afterwards, we reveal that sum-of-power mapping $\Psi_K(\boldsymbol{X})$ has a continuous inverse. Before stating the formal argument, we formally define the injectivity of permutation-invariant mappings:

**Definition 2.7** (Injectivity). A set function $h : \mathbb{R}^{N \times D} \to \mathbb{R}^L$ is injective if there exists a function $g : \mathbb{R}^L \to \mathbb{R}^{N \times D}$ such that for any $\boldsymbol{X} \in \mathbb{R}^{N \times D}$, we have $g \circ f(\boldsymbol{X}) \sim \boldsymbol{X}$. Then $g$ is an inverse of $f$.

And we summarize the existence of continuous inverse of $\Psi_K(\boldsymbol{x})$ into the following lemma shown by [9] and improved by [23]. This result comes from homeomorphism between roots and coefficients of monic polynomials [35].

**Lemma 2.8** (Existence of Continuous Inverse of Sum-of-Power [9,23]). $\Psi_N : \mathbb{R}^N \to \mathbb{R}^N$ *is injective, thus the inverse* $\Psi_N^{-1} : \mathbb{R}^N \to \mathbb{R}^N$ *exists. Moreover,* $\Psi_N^{-1}$ *is continuous.*

Now we are ready to prove necessity in Theorem 2.4 as sufficiency is easy to check. By choosing $\phi = \psi_N : \mathbb{R} \to \mathbb{R}^N$ to be the power mapping (cf. Definition 2.6), and $\rho = f \circ \Psi_N^{-1}$. For any scalar-valued set $\boldsymbol{X} = \begin{bmatrix} x^{(1)} & \cdots & x^{(N)} \end{bmatrix}^\top$, $\rho\left(\sum_{i=1}^{N} \phi(x^{(i)})\right) = f \circ \Psi_N^{-1} \circ \Psi_N(\boldsymbol{x}) = f(\boldsymbol{P}\boldsymbol{X}) = f(\boldsymbol{X})$ for some $\boldsymbol{P} \in \Pi(N)$. The existence and continuity of $\Psi_N^{-1}$ are due to Lemma 2.8.

Theorem 2.4 gives the *exact decomposable form* [23] for permutation-invariant functions, which is stricter than approximation error based expressiveness analysis. In summary, the key idea is to establish a mapping $\phi$ whose element-wise sum-pooling has a continuous inverse.

**Curse of High-dimensional Features.** We argue that the proof of Theorem 2.4 is not applicable to high-dimensional set features ($D \geq 2$). The main reason is that power mapping defined in Definition 2.6 only receives scalar input. It remains elusive how to extend it to a multivariate version that admits injectivity and a continuous inverse. A plausible idea seems to be applying power mapping for each channel $\boldsymbol{x}_i$ independently, and due to the injectivity of sum-of-power mapping $\Psi_N$, each channel can be uniquely recovered individually via the inverse $\Psi_N^{-1}$. However, we point out that each recovered feature channel $\boldsymbol{x}'_i \sim \boldsymbol{x}_i, \forall i \in [D]$, does not imply $\begin{bmatrix} \boldsymbol{x}'_1 & \cdots & \boldsymbol{x}'_D \end{bmatrix} \sim \boldsymbol{X}$, where the alignment of features across channels gets lost. Hence, channel-wise power encoding no more composes an injective mapping. Zaheer et al. [9] proposed to adopt multivariate polynomials as $\phi$ for high-dimensional case, which leverages the fact that multivariate symmetric polynomials are dense in the space of permutation invariant functions (akin to Stone-Wasserstein theorem) [30]. This idea later got formalized in [25] by setting $\phi(\boldsymbol{x}^{(i)}) = \begin{bmatrix} \cdots & \prod_{j \in [D]} (x_j^{(i)})^{\alpha_j} & \cdots \end{bmatrix}$ where $\boldsymbol{\alpha} \in \mathbb{N}^D$ traverses all $\sum_{j \in [D]} \alpha_j \leq n$ and extended to permutation equivariant functions. Nevertheless, the dimension $L = \binom{N+D}{D}$, i.e., exponential in $\min\{N, D\}$ in this case, and unlike DeepSets [9] which exactly recovers $f$ for $D = 1$, the architecture in [9, 25] can only approximate the desired function.

## 3 Main Results

In this section, we present our main result which extends Theorem 2.4 to high-dimensional features. Our conclusion is that to universally represent a set function on sets of length $N$ and feature dimension

161 $D$ with the DeepSets architecture [9] (Eq. (1)), a dimension $L$ at most polynomial in $N$ and $D$ is
162 needed for expressing the intermediate embedding space.

163 Formally, we summarize our main result in the following theorem.

164 **Theorem 3.1** (The main result). *Suppose $D \geq 2$. For any continuous permutation-invariant function*
165 $f : \mathcal{K}^{N \times D} \to \mathbb{R}$, $\mathcal{K} \subseteq \mathbb{R}$, *there exists two continuous mappings* $\phi : \mathbb{R}^D \to \mathbb{R}^L$ *and* $\rho : \mathbb{R}^L \to \mathbb{R}$
166 *such that for every* $\boldsymbol{X} \in \mathcal{K}^{N \times D}$, $f(\boldsymbol{X}) = \rho\left(\sum_{i=1}^N \phi(\boldsymbol{x}^{(i)})\right)$ *where*

167 • *For some $L \in [N(D+1), N^5 D^2]$ when $\phi$ admits **linear layer + power mapping (LP)** architecture:*

$$\phi(\boldsymbol{x}) = \begin{bmatrix} \psi_N(\boldsymbol{w}_1 \boldsymbol{x})^\top & \cdots & \psi_N(\boldsymbol{w}_K \boldsymbol{x})^\top \end{bmatrix} \tag{2}$$

168 *for some $\boldsymbol{w}_1, \cdots, \boldsymbol{w}_K \in \mathbb{R}^D$, and $K = L/N$.*

169 • *For some $L \in [ND, 2N^2 D^2]$ when $\phi$ admits **logarithm activations + linear layer + exponential***
170 ***activations (LLE)** architecture:*

$$\phi(\boldsymbol{x}) = \begin{bmatrix} \exp(\boldsymbol{w}_1 \log(\boldsymbol{x})) & \cdots & \exp(\boldsymbol{w}_L \log(\boldsymbol{x})) \end{bmatrix} \tag{3}$$

171 *for some $\boldsymbol{w}_1, \cdots, \boldsymbol{w}_L \in \mathbb{R}^D$ and $\mathcal{K} \subseteq \mathbb{R}_{>0}$.*

172 The bounds of $L$ depend on the choice of the architecture of $\phi$, which are illustrated in Fig. 1. In
173 the LP setting, we adopt a linear layer that maps each set element into $K$ dimension. Then we apply
174 a channel-wise power mapping that separately transforms each value in the feature vector into an
175 $N$-order power series, and concatenates all the activations together, resulting in a $KN$ dimension
176 feature. The LP architecture is closer to DeepSets [9] as they share the power mapping as the main
177 component. Theorem 3.1 guarantees the existence of $\rho$ and $\phi$ (in the form of Eq. (2)) which satisfy
178 Eq. (1) without the need to set $K$ larger than $N^4 D^2$ while $K \geq D + 1$ is necessary. Therefore, the
179 total embedding size $L = KN$ is bounded by $N^5 D^2$ above and $N(D + 1)$ below. Note that this
180 lower bound is not trivial as $ND$ is the degree of freedom of the input $\boldsymbol{X}$. No matter how $\boldsymbol{w}_1, ..., \boldsymbol{w}_K$
181 are adopted, one cannot achieve an injective mapping by just using $ND$ dimension.

182 In the LLE architecture, we investigate the utilization of logarithmic and exponential activations in set
183 representation, which are also valid activations to build deep neural networks [36, 37]. Each set entry
184 will be squashed by a element-wise logarithm first, then linearly embedded into an $L$-dimensional
185 space via a group of weights, and finally transformed by an element-wise exponential activation.
186 Essentially, each $\exp(\boldsymbol{w}_i \log(\boldsymbol{x})), i \in [L]$ gives a monomial of $\boldsymbol{x}$. The LLE architecture requires the
187 feature space constrained on the positive orthant to ensure logarithmic operations are feasible. But
188 the advantage is that the upper bound of $L$ is improved to be $2N^2 D^2$. The lower bound $ND$ for
189 the LLE architecture is a trivial bound due to the degree of freedom of the input $\boldsymbol{X}$. Note that the
190 constraint on the positive orthant $\mathbb{R}_{>0}$ is not essential. If we are able to use monomial activations to
191 process a vector $\boldsymbol{x}$ as used in [25, 26], then, the constraint on the positive orthant can be removed.

192 *Remark* 3.2. The bounds in Theorem 3.1 are non-asymptotic. This implies the latent dimensions
193 specified by the corresponding architectures are precisely sufficient for expressing the input.

194 *Remark* 3.3. Unlike $\phi$, the form of $\rho$ cannot be explicitly specified, as it depends on the desired
195 function $f$. The complexity of $\rho$ remains unexplored in this paper, which may be high in practice.

196 **Importance of Continuity.** We argue that the requirements of continuity on $\rho$ and $\phi$ are essential
197 for our discussion. First, practical neural networks can only provably approximate continuous
198 functions [27, 28]. Moreover, set representation without such requirements can be straightforward
199 (but likely meaningless in practice). This is due to the following lemma.

200 **Lemma 3.4** ( [38]). *There exists a discontinuous bijective mapping between $\mathbb{R}^D$ and $\mathbb{R}$ if $D \geq 2$.*

201 By Lemma 3.4, we can define a bijective mapping $r : \mathbb{R}^D \to \mathbb{R}$ which maps the high-dimensional
202 features to scalars, and its inverse exists. Then, the same proof of Theorem 2.4 goes through by
203 letting $\phi = \psi_N \circ r$ and $\rho = f \circ r^{-1} \circ \Psi_N^{-1}$. However, we note both $\rho$ and $\phi$ lose continuity.

204 **Comparison with Prior Arts.** Below we highlight the significance of Theorem 3.1 in contrast
205 to the existing literature. A quick overview is listed in Tab. 1 for illustration. The lower bound
206 in Theorem 3.1 corrects a natural misconception that the degree of freedom (i.e., $L = ND$ for

multi-channel cases) is not enough for representing the embedding space. Fortunately, the upper bound in Theorem 3.1 shows the complexity of representing vector-valued sets is still manageable as it merely scales polynomially in $N$ and $D$. Compared with Zweig and Bruna's finding [26], our result significantly improves this bound on $L$ from exponential to polynomial by allowing non-analytic functions to amortize the expressiveness. Besides, Zweig and Bruna's work [26] is hard to be applied to the real domain, while ours are extensible to complex numbers and equivariant functions.

# 4 Proof Sketch

In this section, we introduce the proof techniques of Theorem 3.1, while deferring a full version and all missing proofs to the supplementary materials.

The proof of Theorem 3.1 mainly consists of two steps below, which is completely constructive:

1. For the LP architecture, we construct a group of $K$ linear weights $\boldsymbol{w}_1 \cdots, \boldsymbol{w}_K$ with $K \leq N^4 D^2$ such that the summation over the associated LP embedding (Eq. (2)): $\Psi(\boldsymbol{X}) = \sum_{i=1}^N \phi(\boldsymbol{x}^{(i)})$ is injective and has a continuous inverse. Moreover, if $K \leq D$, such weights do not exist, which induces the lower bound.

2. Similarly, for the LLE architecture, we construct a group of $L$ linear weights $\boldsymbol{w}_1 \cdots, \boldsymbol{w}_L$ with $L \leq 2N^2 D^2$ such that the summation over the associated LLE embedding (Eq. (3)) is injective and has a continuous inverse. Trivially, if $L < ND$, such weights do not exist, which induces the lower bound.

3. Then the proof of upper bounds can be concluded for both settings by letting $\rho = f \circ \Psi^{-1}$ since $\rho \left( \sum_{i=1}^N \phi(\boldsymbol{x}^{(i)}) \right) = f \circ \Psi^{-1} \circ \Psi(\boldsymbol{X}) = f(\boldsymbol{P}\boldsymbol{X}) = f(\boldsymbol{X})$ for some $\boldsymbol{P} \in \Pi(N)$.

Next, we elaborate on the construction idea which yields injectivity for both embedding layers in Sec. 4.1 and 4.2, respectively. To show injectivity, it is equivalent to establish the following statement for both Eq. (2) and Eq. (3), respectively:

$$\forall \boldsymbol{X}, \boldsymbol{X}' \in \mathbb{R}^{N \times D}, \sum_{i=1}^N \phi(\boldsymbol{x}^{(i)}) = \sum_{i=1}^N \phi(\boldsymbol{x}'^{(i)}) \Rightarrow \boldsymbol{X} \sim \boldsymbol{X}' \tag{4}$$

In Sec. 4.3, we prove the continuity of the inverse map for LP and LLE via arguments similar to [35].

## 4.1 Injectivity of LP

In this section, we consider $\phi$ follows the definition in Eq. (2), which amounts to first linearly transforming each set element and then applying channel-wise power mapping. This is, we seek a group of linear transformations $\boldsymbol{w}_1, \cdots, \boldsymbol{w}_K$ such that $\boldsymbol{X} \sim \boldsymbol{X}'$ can be induced from $\boldsymbol{X}\boldsymbol{w}_i \sim \boldsymbol{X}'\boldsymbol{w}_i, \forall i \in [K]$ for some $K$ larger than $N$ while being polynomial in $N$ and $D$. The intuition is that linear mixing among each channel can encode relative positional information. Only if $\boldsymbol{X} \sim \boldsymbol{X}'$, the mixing information can be reproduced.

Formally, the first step accords to the property of power mapping (cf. Lemma 2.8), and we can obtain:

$$\sum_{i=1}^N \phi(\boldsymbol{x}^{(i)}) = \sum_{i=1}^N \phi(\boldsymbol{x}'^{(i)}) \Rightarrow \boldsymbol{X}\boldsymbol{w}_i \sim \boldsymbol{X}'\boldsymbol{w}_i, \forall i \in [K]. \tag{5}$$

To induce $\boldsymbol{X} \sim \boldsymbol{X}'$ from $\boldsymbol{X}\boldsymbol{w}_i \sim \boldsymbol{X}'\boldsymbol{w}_i, \forall i \in [K]$, our construction divides the weights $\{\boldsymbol{w}_i, i \in [K]\}$ into three groups: $\{\boldsymbol{w}_i^{(1)} : i \in [D]\}$, $\{\boldsymbol{w}_j^{(2)} : j \in [K_1]\}$, and $\{\boldsymbol{w}_{i,j,k}^{(3)} : i \in [D], j \in [K_1], k \in [K_2]\}$. Each block is outlined as below:

1. Let the first group of weights $\boldsymbol{w}_1^{(1)} = \boldsymbol{e}_1, \cdots, \boldsymbol{w}_D^{(1)} = \boldsymbol{e}_D$ to buffer the original features, where $\boldsymbol{e}_i$ is the $i$-th canonical basis.

2. Design the second group of linear weights, $\boldsymbol{w}_1^{(2)}, \cdots, \boldsymbol{w}_{K_1}^{(2)}$ for $K_1$ as large as $N(N-1)(D-1)/2 + 1$, which, by Lemma 4.4 latter, guarantees at least one of $\boldsymbol{X}\boldsymbol{w}_j^{(2)}, j \in [K_1]$ forms an anchor defined below:

**Definition 4.1** (Anchor). Consider the data matrix $\boldsymbol{X} \in \mathbb{R}^{N \times D}$, then $\boldsymbol{a} \in \mathbb{R}^N$ is called an anchor of $\boldsymbol{X}$ if $\boldsymbol{a}_i \neq \boldsymbol{a}_j$ for any $i, j \in [N]$ such that $\boldsymbol{x}^{(i)} \neq \boldsymbol{x}^{(j)}$.

And suppose $\boldsymbol{a} = \boldsymbol{X}\boldsymbol{w}_{j^*}^{(2)}$ is an anchor of $\boldsymbol{X}$ for some $j^* \in [K_1]$ and $\boldsymbol{a}' = \boldsymbol{X}'\boldsymbol{w}_{j^*}^{(2)}$, then we show the following statement is true by Lemma 4.3 latter:

$$[\boldsymbol{a} \quad \boldsymbol{x}_i] \sim [\boldsymbol{a}' \quad \boldsymbol{x}'_i], \forall i \in [D] \Rightarrow \boldsymbol{X} \sim \boldsymbol{X}'. \tag{6}$$

3. Design a group of weights $\boldsymbol{w}_{i,j,k}^{(3)}$ for $i \in [D], j \in [K_1], k \in [K_2]$ with $K_2 = N(N-1) + 1$ that mixes each original channel $\boldsymbol{x}_i$ with each $\boldsymbol{X}\boldsymbol{w}_j^{(2)}, j \in [K_1]$ by $\boldsymbol{w}_{i,j,k}^{(3)} = \boldsymbol{e}_i - \gamma_k \boldsymbol{w}_j^{(2)}$. Then we show in Lemma 4.5 that:

$$\boldsymbol{X}\boldsymbol{w}_i \sim \boldsymbol{X}'\boldsymbol{w}_i, \forall i \in [K] \Rightarrow \left[\boldsymbol{X}\boldsymbol{w}_j^{(2)} \quad \boldsymbol{x}_i\right] \sim \left[\boldsymbol{X}'\boldsymbol{w}_j^{(2)} \quad \boldsymbol{x}'_i\right], \forall i \in [D], j \in [K_1] \tag{7}$$

With such configuration, injectivity can be concluded by the entailment along Eq. (5), (7), (6): Eq. (5) guarantees the RHS of Eq. (7); The existence of the anchor in Lemma 4.4 paired with Eq. (6) guarantees $\boldsymbol{X} \sim \boldsymbol{X}'$. The total required number of weights $K = D + K_1 + DK_1K_2 \leq N^4D^2$.

Below we provides a series of lemmas that demonstrate the desirable properties of anchors and elaborate on the construction complexity. Detailed proofs are left in Appendix. In plain language, by Definition 4.1, two entries in the anchor must be distinctive if the set elements at the corresponding indices are not equal. As a consequence, we derive the following property of anchors:

**Lemma 4.2.** *Consider the data matrix $\boldsymbol{X} \in \mathbb{R}^{N \times D}$ and $\boldsymbol{a} \in \mathbb{R}^N$ an anchor of $\boldsymbol{X}$. Then if there exists $\boldsymbol{P} \in \Pi(N)$ such that $\boldsymbol{P}\boldsymbol{a} = \boldsymbol{a}$ then $\boldsymbol{P}\boldsymbol{x}_i = \boldsymbol{x}_i$ for every $i \in [D]$.*

With the above property, anchors defined in Definition 4.1 indeed have the entailment in Eq. (6):

**Lemma 4.3** (Union Alignment based on Anchor Alignment). *Consider the data matrix $\boldsymbol{X}, \boldsymbol{X}' \in \mathbb{R}^{N \times D}$, $\boldsymbol{a} \in \mathbb{R}^N$ is an anchor of $\boldsymbol{X}$ and $\boldsymbol{a}' \in \mathbb{R}^N$ is an arbitrary vector. If $[\boldsymbol{a} \quad \boldsymbol{x}_i] \sim [\boldsymbol{a}' \quad \boldsymbol{x}'_i]$ for every $i \in [D]$, then $\boldsymbol{X} \sim \boldsymbol{X}'$.*

However, the anchor $\boldsymbol{a}$ is required to be generated from $\boldsymbol{X}$ via a point-wise linear transformation. The strategy to generate an anchor is to enumerate as many linear weights as needs, so that for any $\boldsymbol{X}$, at least one $j$ such that $\boldsymbol{X}\boldsymbol{w}_j^{(2)}$ becomes an anchor. We show that at most $N(N-1)(D-1)/2 + 1$ linear weights are enough to guarantee the existence of an anchor for any $\boldsymbol{X}$:

**Lemma 4.4** (Anchor Construction). *There exists a set of weights $\boldsymbol{w}_1, \cdots, \boldsymbol{w}_K$ where $K = N(N-1)(D-1)/2 + 1$ such that for every data matrix $\boldsymbol{X} \in \mathbb{R}^{N \times D}$, there exists $j \in [K]$, $\boldsymbol{X}\boldsymbol{w}_j$ is an anchor of $\boldsymbol{X}$.*

We wrap off the proof by presenting the following lemma which is applied to prove Eq. (7) by fixing any $i \in [D], j \in [K_1]$ in Eq. (7) while checking the condition for all $k \in [K_2]$:

**Lemma 4.5** (Anchor Matching). *There exists a group of coefficients $\gamma_1, \cdots, \gamma_{K_2}$ where $K_2 = N(N-1) + 1$ such that the following statement holds: Given any $\boldsymbol{x}, \boldsymbol{x}', \boldsymbol{y}, \boldsymbol{y}' \in \mathbb{R}^N$ such that $\boldsymbol{x} \sim \boldsymbol{x}'$ and $\boldsymbol{y} \sim \boldsymbol{y}'$, if $(\boldsymbol{x} - \gamma_k \boldsymbol{y}) \sim (\boldsymbol{x}' - \gamma_k \boldsymbol{y}')$ for every $k \in [K_2]$, then $[\boldsymbol{x} \quad \boldsymbol{y}] \sim [\boldsymbol{x}' \quad \boldsymbol{y}']$.*

For completeness, we add the following lemma which implies LP-induced sum-pooling cannot be injective if $K \leq ND$, when $D \geq 2$.

**Theorem 4.6** (Lower Bound). *Consider data matrices $\boldsymbol{X} \in \mathbb{R}^{N \times D}$ where $D \geq 2$. If $K \leq D$, then for every $\boldsymbol{w}_1, \cdots, \boldsymbol{w}_K$, there exists $\boldsymbol{X}' \in \mathbb{R}^{N \times D}$ such that $\boldsymbol{X} \not\sim \boldsymbol{X}'$ but $\boldsymbol{X}\boldsymbol{w}_i \sim \boldsymbol{X}'\boldsymbol{w}_i$ for every $i \in [K]$.*

*Remark* 4.7. Theorem 4.6 is significant in that with high-dimensional features, the injectivity is provably not satisfied when the embedding space has dimension equal to the degree of freedom.

## 4.2 Injectivity of LLE

In this section, we consider $\phi$ follows the definition in Eq. (3). First of all, we note that each term in the RHS of Eq. (3) can be rewritten as a monomial as shown in Eq. (8). Suppose we are able to use monomial activations to process a vector $\boldsymbol{x}^{(i)}$. Then, the constraint on the positive orthant $\mathbb{R}_{>0}$ in our main result Theorem 3.1 can be even removed.

$$\phi(\boldsymbol{x}) = [\cdots \quad \exp(\boldsymbol{w}_i \log(\boldsymbol{x})) \quad \cdots] = \left[\cdots \quad \prod_{j=1}^D \boldsymbol{x}_j^{\boldsymbol{w}_{i,j}} \quad \cdots\right] \tag{8}$$

291 Then, the assignment of $\boldsymbol{w}_1, \cdots, \boldsymbol{w}_L$ amounts to specifying the exponents for $D$ power functions
292 within the product. Next, we prepare our construction with the following two lemmas:

**Lemma 4.8.** *For any pair of vectors* $\boldsymbol{x}_1, \boldsymbol{x}_2 \in \mathbb{R}^N, \boldsymbol{y}_1, \boldsymbol{y}_2 \in \mathbb{R}^N$, *if* $\sum_{i \in [N]} \boldsymbol{x}_{1,i}^{l-k} \boldsymbol{x}_{2,i}^k =$
$\sum_{i \in [N]} \boldsymbol{y}_{1,i}^{l-k} \boldsymbol{y}_{2,i}^k$ *for every* $l, k \in [N]$ *such that* $0 \le k \le l$, *then* $[\boldsymbol{x}_1 \quad \boldsymbol{x}_2] \sim [\boldsymbol{y}_1 \quad \boldsymbol{y}_2]$.

295 The above lemma is to show that we may use summations of monic bivariate monomials to align every
296 two feature columns. The next lemma shows that such pairwise alignment yields union alignment.

**Lemma 4.9** (Union Alignment based on Pairwise Alignment). *Consider data matrices* $\boldsymbol{X}, \boldsymbol{X}' \in$
$\mathbb{R}^{N \times D}$. *If* $[\boldsymbol{x}_i \quad \boldsymbol{x}_j] \sim [\boldsymbol{x}'_i \quad \boldsymbol{x}'_j]$ *for every* $i, j \in [D]$, *then* $\boldsymbol{X} \sim \boldsymbol{X}'$.

299 Then the construction idea of $\boldsymbol{w}_1, \cdots, \boldsymbol{w}_L$ can be drawn from Lemma 4.8 and 4.9:

300 1. Lemma 4.8 indicates if the weights in Eq. (8) enumerate all the monic bivariate monomials in
301  each pair of channels with degrees less or equal to $N$, i.e., $\boldsymbol{x}_i^p \boldsymbol{x}_j^q$ for all $i, j \in [D]$ and $p + q \le N$,
302  then we can yield:

$$\sum_{i=1}^N \phi(\boldsymbol{x}^{(i)}) = \sum_{i=1}^N \phi(\boldsymbol{x}'^{(i)}) \Rightarrow [\boldsymbol{x}_i \quad \boldsymbol{x}_j] \sim [\boldsymbol{x}'_i \quad \boldsymbol{x}'_j], \forall i, j \in [D]. \tag{9}$$

303 2. The next step is to invoke Lemma 4.9 which implies if every pair of feature channels is aligned,
304  then we can conclude all the channels are aligned with each other as well.

$$[\boldsymbol{x}_i \quad \boldsymbol{x}_j] \sim [\boldsymbol{x}'_i \quad \boldsymbol{x}'_j], \forall i, j \in [D] \Rightarrow \boldsymbol{X} \sim \boldsymbol{X}'. \tag{10}$$

305 Based on these motivations, we assign the weights that induce all bivariate monic monomials with
306 the degree no more than $N$. First of all, we reindex $\{\boldsymbol{w}_i, i \in [L]\}$ as $\{\boldsymbol{w}_{i,j,p,q}, i \in [D], j \in [D], p \in$
307 $[N], q \in [p+1]\}$. Then weights can be explicitly specified as $\boldsymbol{w}_{i,j,p,q} = (q-1)\boldsymbol{e}_i + (p-q+1)\boldsymbol{e}_j$,
308 where $\boldsymbol{e}_i$ is the $i$-th canonical basis. With such weights, injectivity can be concluded by entailment
309 along Eq. (9) and (10). Moreover, the total number of linear weights is $L = D^2(N+3)N/2 \le$
310 $2N^2 D^2$, as desired.

### 4.3 Continuous Lemma

312 In this section, we show that the LP and LLE induced sum-pooling are both homeomorphic. We
313 note that it is intractable to obtain the closed form of their inverse maps. Notably, the following
314 remarkable result can get rid of inversing a functions explicitly by merely examining the topological
315 relationship between the domain and image space.

**Lemma 4.10.** *(Theorem 1.2 [35]) Let* $(\mathcal{X}, d_{\mathcal{X}})$ *and* $(\mathcal{Y}, d_{\mathcal{Y}})$ *be two metric spaces and* $f : \mathcal{X} \to \mathcal{Y}$ *is*
317 *a bijection such that (a) each bounded and closed subset of* $\mathcal{X}$ *is compact, (b)* $f$ *is continuous, (c)*
318 $f^{-1}$ *maps each bounded set in* $\mathcal{Y}$ *into a bounded set in* $\mathcal{X}$. *Then* $f^{-1}$ *is continuous.*

319 Subsequently, we show the continuity in an informal but more intuitive way while deferring a rigorous
320 version to the supplementary materials. Denote $\Psi(\boldsymbol{X}) = \sum_{i \in [N]} \phi(\boldsymbol{x}^{(i)})$. To begin with, we set
321 $\mathcal{X} = \mathbb{R}^{N \times D}/ \sim$ with metric $d_{\mathcal{X}}(\boldsymbol{X}, \boldsymbol{X}') = \min_{\boldsymbol{P} \in \Pi(N)} \|\boldsymbol{X} - \boldsymbol{P}\boldsymbol{X}'\|_1$ and $\mathcal{Y} = \{\Psi(\boldsymbol{X}) | \boldsymbol{X} \in$
322 $\mathcal{X}\} \subseteq \mathbb{R}^L$ with metric $d_{\mathcal{Y}}(\boldsymbol{y}, \boldsymbol{y}') = \|\boldsymbol{y} - \boldsymbol{y}'\|_\infty$. It is easy to show that $\mathcal{X}$ satisfies the conditions
323 (a) and $\Psi(\boldsymbol{X})$ satisfies (b) for both LP and LLE embedding layers. Then it remains to conclude the
324 proof by verifying the condition (c) for the mapping $\mathcal{Y} \to \mathcal{X}$, i.e., the inverse of $\Psi(\boldsymbol{X})$. We visualize
325 this mapping following our arguments on injectivity:

$$
\begin{array}{cccc}
(LP) & \Psi(\boldsymbol{X}) \xrightarrow{\text{Eq. (5)}} & [\cdots \quad \boldsymbol{P}_i\boldsymbol{X}\boldsymbol{w}_i \quad \cdots], i \in [K] & \xrightarrow{\text{Eqs. (6) + (7)}} \boldsymbol{P}\boldsymbol{X} \\
(LLE) & \underbrace{\Psi(\boldsymbol{X})}_{\mathcal{Y}} \xrightarrow{\text{Eq. (9)}} & \underbrace{[\cdots \quad \boldsymbol{Q}_{i,j}\boldsymbol{x}_i \quad \boldsymbol{Q}_{i,j}\boldsymbol{x}_j \quad \cdots], i, j \in [D]}_{\mathcal{Z}} & \xrightarrow{\text{Eq. (10)}} \underbrace{\boldsymbol{Q}\boldsymbol{X}}_{\mathcal{X}} ,
\end{array}
$$

326 for some $\boldsymbol{X}$ dependent $\boldsymbol{P}, \boldsymbol{Q}$. Here, $\boldsymbol{P}_i, i \in [K]$ and $\boldsymbol{Q}_{i,j}, i, j \in [D] \in \Pi(N)$. According to
327 homeomorphism between polynomial coefficients and roots (Theorem 3.4 in [35]), any bounded set
328 in $\mathcal{Y}$ will induce a bound set in $\mathcal{Z}$. Moreover, since elements in $\mathcal{Z}$ contains all the columns of $\mathcal{X}$ (up
329 to some changes of the entry orders), a bounded set in $\mathcal{Z}$ also corresponds to a bounded set in $\mathcal{X}$.
330 Through this line of arguments, we conclude the proof.

## 5 Extensions

In this section, we discuss two extensions to Theorem 3.1, which strengthen our main result.

**Permutation Equivariance.** Permutation-equivariant functions (cf. Definition 2.3) are considered as a more general family of set functions. Our main result does not lose generality to this class of functions. By Lemma 2 of [7], Theorem 3.1 can be directly extended to permutation-equivariant functions with *the same lower and upper bounds*, stated as follows:

**Theorem 5.1** (Extension to Equivariance). *For any permutation-equivariant function* $f : \mathcal{K}^{N \times D} \to \mathbb{R}^N$, $\mathcal{K} \subseteq \mathbb{R}$, *there exists continuous functions* $\phi : \mathbb{R}^D \to \mathbb{R}^L$ *and* $\rho : \mathbb{R}^D \times \mathbb{R}^L \to \mathbb{R}$ *such that*

$$f(\boldsymbol{X})_j = \rho\left(\boldsymbol{x}^{(j)}, \sum_{i \in [N]} \phi(\boldsymbol{x}^{(i)})\right) \text{ for every } j \in [N], \text{ where } L \in [N(D+1), N^5 D^2] \text{ when } \phi$$

*admits LP architecture, and* $L \in [ND, 2N^2 D^2]$ *when* $\phi$ *admits LLE architecture* $(\mathcal{K} \in \mathbb{R}_{>0})$.

**Complex Domain.** The upper bounds in Theorem 3.1 is also true to complex features up to a constant scale (i.e., $\mathcal{K} \subseteq \mathbb{C}$). When features are defined over $\mathbb{C}^{N \times D}$, our primary idea is to divide each channel into two real feature vectors, and recall Theorem 3.1 to conclude the arguments on an $\mathbb{R}^{N \times 2D}$ input. All of our proof strategies are still applied. This result directly contrasts to Zweig and Bruna's work [26] whose main arguments were established on complex numbers. We show that even moving to the complex domain, polynomial length of $L$ is still sufficient for the DeepSets architecture [9]. We state a formal version of the theorem in the supplementary material.

## 6 Related Work

Works on neural networks to represent set functions have been discussed extensively in the Sec. 1. Here, we review other related works on the expressive power analysis of neural networks.

Early works studied the expressive power of feed-forward neural networks with different activations [27, 28]. Recent works focused on characterizing the benefits of the expressive power of deep architectures to explain their empirical success [39–43]. Modern neural networks often enforce some invariance properties into their architectures such as CNNs that capture spatial translation invariance. The expressive power of invariant neural networks has been analyzed recently [22, 44, 45].

The architectures studied in the above works allow universal approximation of continuous functions defined on their inputs. However, the family of practically useful architectures that enforce permutation invariance often fail in achieving universal approximation. Graph Neural Networks (GNNs) enforce permutation invariance and can be viewed as an extension of set neural networks to encode a set of pair-wise relations instead of a set of individual elements [20, 21, 46, 47]. GNNs suffer from limited expressive power [5, 17, 18] unless they adopt exponential-order tensors [48]. Hence, previous studies often characterized GNNs' expressive power based on their capability of distinguishing non-isomorphic graphs. Only a few works have ever discussed the function approximation property of GNNs [49–51] while these works still miss characterizing such dependence on the depth and width of the architectures [52]. As practical GNNs commonly adopt the architectures that combine feed-forward neural networks with set operations (neighborhood aggregation), we believe the characterization of the needed size for set function approximation studied in [26] and this work may provide useful tools to study finer-grained characterizations of the expressive power of GNNs.

## 7 Conclusion

This work investigates how many neurons are needed to model the embedding space for set representation learning with the DeepSets architecture [9]. Our paper provides an affirmative answer that polynomial many neurons in the set size and feature dimension are sufficient. Compared with prior arts, our theory takes high-dimensional features into consideration while significantly advancing the state-of-the-art results from exponential to polynomial.

**Limitations.** The tightness of our bounds is not examined in this paper, and the complexity of $\rho$ is uninvestigated and left for future exploration. Besides, deriving an embedding layer agnostic lower bound for the embedding space remains another widely open question.

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
