$$\begin{bmatrix} \boldsymbol{a} & \boldsymbol{x}_i \end{bmatrix} \sim \begin{bmatrix} \boldsymbol{a}' & \boldsymbol{x}'_i \end{bmatrix}, \forall i \in [D] \Rightarrow \boldsymbol{X} \sim \boldsymbol{X}'. \tag{6}$$

3. Design a group of weights $\boldsymbol{w}_{i,j,k}^{(3)}$ for $i \in [D], j \in [K_1], k \in [K_2]$ with $K_2 = N(N-1) + 1$ that mixes each original channel $\boldsymbol{x}_i$ with each $\boldsymbol{X}\boldsymbol{w}_j^{(2)}, j \in [K_1]$ by $\boldsymbol{w}_{i,j,k}^{(3)} = \boldsymbol{e}_i - \gamma_k \boldsymbol{w}_j^{(2)}$. Then we show in Lemma 4.5 that:

$$\boldsymbol{X}\boldsymbol{w}_i \sim \boldsymbol{X}'\boldsymbol{w}_i, \forall i \in [K] \Rightarrow \begin{bmatrix} \boldsymbol{X}\boldsymbol{w}_j^{(2)} & \boldsymbol{x}_i \end{bmatrix} \sim \begin{bmatrix} \boldsymbol{X}'\boldsymbol{w}_j^{(2)} & \boldsymbol{x}'_i \end{bmatrix}, \forall i \in [D], j \in [K_1] \tag{7}$$

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

## A   Formal Definitions

In this section, we begin by providing rigorous definitions to specify the topology of the input space of permutation-invariant functions.

**Definition A.1.** Equipped $\mathcal{K}^{N \times D}$ with the equivalence relation $\sim$ (cf. Definition 2.1), we define metric space $(\mathcal{K}^{N \times D} / \sim, d_F)$, where $d_F : (\mathcal{K}^{N \times D} / \sim) \times (\mathcal{K}^{N \times D} / \sim) \to \mathbb{R}_{\geq 0}$ is the optimal transport distance:

$$d_F(\boldsymbol{X}, \boldsymbol{X}') = \min_{\boldsymbol{P} \in \Pi(N)} \|\boldsymbol{P}\boldsymbol{X} - \boldsymbol{X}'\|_{\infty,\infty}, \tag{11}$$

and $\mathcal{K}$ can be either $\mathbb{R}$ or $\mathbb{C}$.

*Remark* A.2. The $\|\cdot\|_{\infty,\infty}$ norm takes the absolute value of the maximal entry: $\max_{i \in [N], j \in [D]} |\boldsymbol{X}_{i,j}|$. Other topologically equivalent matrix norms also apply.

**Lemma A.3.** *The function* $d_F : (\mathcal{K}^{N \times D} / \sim) \times (\mathcal{K}^{N \times D} / \sim) \to \mathbb{R}_{\geq 0}$ *is a distance metric on* $\mathcal{K}^{N \times D} / \sim$.

*Proof.* Identity, positivity, and symmetry trivially hold for $d_F$. It remains to show the triangle inequality as below: for arbitrary $\boldsymbol{X}, \boldsymbol{X}', \boldsymbol{X}'' \in (\mathcal{K}^{N \times D} / \sim, d_F)$,

$$\begin{aligned} d_F(\boldsymbol{X}, \boldsymbol{X}'') = \min_{\boldsymbol{P} \in \Pi(N)} \|\boldsymbol{P}\boldsymbol{X} - \boldsymbol{X}''\|_{\infty,\infty} &\leq \min_{\boldsymbol{P} \in \Pi(N)} \left( \|\boldsymbol{P}\boldsymbol{X} - \boldsymbol{Q}^*\boldsymbol{X}'\|_{\infty,\infty} + \|\boldsymbol{Q}^*\boldsymbol{X}' - \boldsymbol{X}''\|_{\infty,\infty} \right) \\ &= \min_{\boldsymbol{P} \in \Pi(N)} \|\boldsymbol{P}\boldsymbol{X} - \boldsymbol{Q}^*\boldsymbol{X}'\|_{\infty,\infty} + \|\boldsymbol{Q}^*\boldsymbol{X}' - \boldsymbol{X}''\|_{\infty,\infty} \\ &= d_F(\boldsymbol{X}, \boldsymbol{X}') + d_F(\boldsymbol{X}, \boldsymbol{X}''), \end{aligned}$$

where $\boldsymbol{Q}^* = \operatorname{argmin}_{\boldsymbol{Q} \in \Pi(N)} \|\boldsymbol{Q}\boldsymbol{X}' - \boldsymbol{X}''\|_{\infty,\infty}$. $\qquad\square$

Also we reveal a topological property for $(\mathcal{K}^{N \times D} / \sim, d_F)$ which is essential to show continuity later.

**Lemma A.4.** *Each bounded and closed subset of* $(\mathcal{K}^{N \times D} / \sim, d_F)$ *is compact.*

*Proof.* Without loss of generality, the proof is done by extending Theorem 2.4 in [35] to high-dimensional set elements. $\qquad\square$

Then we can rephrase the definition of permutation invariant function as a proper function mapping between the two metric spaces: $f : (\mathcal{K}^{N \times D} / \sim, d_F) \to (\mathcal{K}^{D'}, d_\infty)$, where $d_\infty : \mathcal{K}^{D'} \times \mathcal{K}^{D'} \to \mathbb{R}_{\geq 0}$ is the $\ell_\infty$-norm induced distance metric.

We also recall the definition of injectivity for permutation-invariant functions:

**Definition A.5** (Injectivity)**.** The following statements are equivalent:

1. A permutation-invariant function $f : (\mathcal{K}^{N \times D} / \sim, d_F) \to (\mathcal{K}^{D'}, d_\infty)$ is injective.

2. There exists a function $g : (\mathcal{K}^{D'}, d_\infty) \to (\mathcal{K}^{N \times D} / \sim, d_F)$ such that for every $\boldsymbol{X} \in \mathcal{K}^{N \times D}$, $g \circ f(\boldsymbol{X}) \sim \boldsymbol{X}$.

3. For every $\boldsymbol{X}, \boldsymbol{X}' \in \mathcal{K}^{N \times D}$ such that $f(\boldsymbol{X}) = f(\boldsymbol{X}')$, then $\boldsymbol{X} \sim \boldsymbol{X}'$.

We give an intuitive definition of continuity for permutation-invariant functions via the epsilon-delta statement:

**Definition A.6** (Continuity)**.** A permutation-invariant function $f : (\mathcal{K}^{N \times D} / \sim, d_F) \to (\mathcal{K}, d_\infty)$ is continuous if for arbitrary $\boldsymbol{X} \in \mathcal{K}^{N \times D}$ and $\epsilon > 0$, there exists $\delta > 0$ such that for every $\boldsymbol{X}' \in \mathcal{K}^{N \times D}$, $d_F(\boldsymbol{X}, \boldsymbol{X}') < \delta$ then $d_\infty(f(\boldsymbol{X}) - f(\boldsymbol{X}')) < \epsilon$.

*Remark* A.7. Since $d_F$ is a distance metric, other equivalent definitions of continuity still applies.

## B  Sum-of-Power Mapping

In this section, we extend Definition 2.6 and Lemma 2.8 to the complex version, which provides the mathematical tools for our later proof.

**Definition B.1.** Define (complex) power mapping: $\psi_N : \mathbb{C} \to \mathbb{C}^N$, $\psi_N(z) = \begin{bmatrix} z & z^2 & \cdots & z^N \end{bmatrix}^\top$ and (complex) sum-of-power mapping $\psi_N : (\mathbb{C}^N / \sim, d_F) \to (\mathbb{C}^N, d_\infty)$, $\psi_N(\boldsymbol{z}) = \sum_{i=1}^N \psi_N(z_i)$.

**Lemma B.2** (Existence of Continuous Inverse of Complex Sum-of-Power [35]). *$\psi_N$ is injective, thus the inverse $\psi_N^{-1} : (\mathbb{C}^N, d_\infty) \to (\mathbb{C}^N / \sim, d_F)$ exists. Moreover, $\psi_N^{-1}$ is continuous.*

**Lemma B.3** (Corollary 3.2 [35]). *Consider a function $\zeta : (\mathbb{C}^N, d_\infty) \to (\mathbb{C}^N / \sim, d_F)$ that maps the coefficients of a polynomial to its root multi-set. Then for any bounded subset $\mathcal{U} \subset (\mathbb{C}^N, d_\infty)$, the image $\zeta(\mathcal{U}) = \{\zeta(\boldsymbol{z}) : \boldsymbol{z} \in \mathcal{U}\}$ is also bounded.*

*Remark B.4.* Lemma B.3 is also true for real numbers when we constrain the domain of $\zeta$ to be real coefficients such that the corresponding polynomial can fully split over the real domain.

**Lemma B.5.** *Consider the $N$-degree sum-of-power mapping: $\psi_N : (\mathbb{C}^{\mathbb{C},N} / \sim, d_F) \to (\mathbb{C}^N, d_\infty)$, where $\psi_N(\boldsymbol{x}) = \sum_{i=1}^N \psi_N(x_i)$. Denote the range of $\psi_N$ as $\mathcal{Z} \subseteq \mathbb{C}^N$ and its inverse mapping $\psi_N^{-1} : (\mathcal{Z}, d_\infty) \to (\mathbb{C}^N / \sim, d_F)$ (existence guaranteed by Lemma B.2). Then for every bounded set $\mathcal{U} \subset (\mathcal{Z}, d_\infty)$, the image $\Psi_N^{-1}(\mathcal{U}) = \{\Psi_N^{-1}(\boldsymbol{z}) : \boldsymbol{z} \in \mathcal{U}\}$ is also bounded.*

*Proof.* We borrow the proof technique from [9] to reveal a polynomial mapping between $(\mathcal{Z}, d_\infty)$ and $(\mathbb{C}^N / \sim, d_F)$. For every $\boldsymbol{\xi} \in (\mathbb{C}^N / \sim, d_F)$, let $\boldsymbol{z} = \psi_N(\boldsymbol{\xi})$ and construct a polynomial:

$$P_{\boldsymbol{\xi}}(x) = \prod_{i=1}^N (x - \xi_i) = x^N - a_1 x^{N-1} + \cdots + (-1)^{N-1} a_{N-1} x + (-1)^N a_N, \qquad (12)$$

where $\boldsymbol{\xi}$ are the roots of $P_{\boldsymbol{\xi}}(x)$ and the coefficients can be written as elementary symmetric polynomials, i.e.,

$$a_n = \sum_{1 \leq j_1 \leq j_2 \leq \cdots \leq j_n \leq N} \xi_{j_1} \xi_{j_2} \cdots \xi_{j_n}, \forall n \in [N]. \qquad (13)$$

On the other hand, the elementary symmetric polynomials can be uniquely expressed as a function of $\boldsymbol{z}$ by Newton-Girard formula:

$$a_n = \frac{1}{n} \det \begin{bmatrix} z_1 & 1 & 0 & 0 & \cdots & 0 \\ z_2 & z_1 & 1 & 0 & \cdots & 0 \\ \vdots & \vdots & \vdots & \vdots & \ddots & \vdots \\ z_{n-1} & z_{n-2} & z_{n-3} & z_{n-4} & \cdots & 1 \\ z_n & z_{n-1} & z_{n-2} & z_{n-3} & \cdots & 1 \end{bmatrix} := Q(\boldsymbol{z}), \forall n \in [N] \qquad (14)$$

where the determinant $Q(\boldsymbol{z})$ is also a polynomial in $\boldsymbol{z}$. Then the proof proceeds by observing that for any bounded subset $\mathcal{U} \subseteq (\mathcal{Z}, d_\infty)$, the resulting $\mathcal{A} = Q(\mathcal{U})$ is also bounded in $(\mathbb{C}^N, d_\infty)$. Therefore, by Lemma B.3, any bounded coefficient set $\mathcal{A}$ will produce a bounded root multi-set in $(\mathbb{C}^N / \sim, d_F)$. $\qquad \square$

*Remark B.6.* Lemma B.3 is also true for real numbers. By Remark B.4, we can constrain the ambient space of $\mathcal{A}$ in Lemma B.3 to be real coefficients whose corresponding polynomial can split over real numbers, and the same proof proceeds.

## C  Proofs of LP Embedding Layer

In this section, we complete the proofs for the LP embedding layer (Eq. (2)). First we constructively show an upper bound that sufficiently achieves injectivity following our discussion in Sec. 4.1, and then prove Theorem 4.6 to reveal a lower bound that is necessary for injectivity. Finally, we show prove the continuity of the inverse of our constructed LP embedding layer with the techniques introduced in Sec. 4.3.

## C.1 Upper Bound for Injectivity

To prove the upper bound, we construct an LP embedding layer with $L \le N^5 D^2$ output neurons such that its induced summation is injective. The main ingredient of our construction is anchor defined in Definition 4.1. Two key properties of anchors are stated in Lemma 4.2 and 4.3 (restated as follows) and proved below:

**Lemma C.1.** *Consider the data matrix $\boldsymbol{X} \in \mathbb{R}^{N \times D}$ and $\boldsymbol{a} \in \mathbb{R}^N$ an anchor of $\boldsymbol{X}$. Then if there exists $\boldsymbol{P} \in \Pi(N)$ such that $\boldsymbol{P}\boldsymbol{a} = \boldsymbol{a}$ then $\boldsymbol{P}\boldsymbol{x}_i = \boldsymbol{x}_i$ for every $i \in [D]$.*

*Proof of Lemma 4.2.* Prove by contradiction. Suppose $\boldsymbol{P}\boldsymbol{x}_i \ne \boldsymbol{x}_i$ for some $i \in [D]$, then there exist some $p, q \in [N]$ such that $x_i^{(p)} \ne x_i^{(q)}$ while $a_p = a_q$. However, this contradicts the definition of an anchor (cf. Definition 4.1). $\square$

**Lemma C.2** (Union Alignment based on Anchor Alignment)**.** *Consider the data matrix $\boldsymbol{X}, \boldsymbol{X}' \in \mathbb{R}^{N \times D}$, $\boldsymbol{a} \in \mathbb{R}^N$ is an anchor of $\boldsymbol{X}$ and $\boldsymbol{a}' \in \mathbb{R}^N$ is an arbitrary vector. If $[\boldsymbol{a} \quad \boldsymbol{x}_i] \sim [\boldsymbol{a}' \quad \boldsymbol{x}'_i]$ for every $i \in [D]$, then $\boldsymbol{X} \sim \boldsymbol{X}'$.*

*Proof of Lemma 4.3.* According to definition of equivalence, there exists $\boldsymbol{Q}_i \in \Pi(N)$ for every $i \in [D]$ such that $[\boldsymbol{a} \quad \boldsymbol{x}_i] = [\boldsymbol{Q}_i \boldsymbol{a}' \quad \boldsymbol{Q}_i \boldsymbol{x}'_i]$. Moreover, since $[\boldsymbol{a} \quad \boldsymbol{x}_i] \sim [\boldsymbol{a}' \quad \boldsymbol{x}'_i]$, it must hold that $\boldsymbol{a} \sim \boldsymbol{a}'$, i.e., there exists $\boldsymbol{P} \in \Pi_N$ such that $\boldsymbol{P}\boldsymbol{a} = \boldsymbol{a}'$. Combined together, we have that $\boldsymbol{Q}_i \boldsymbol{P} \boldsymbol{a} = \boldsymbol{a}$.

Next, we choose $\boldsymbol{Q}'_i = \boldsymbol{P}^\top \boldsymbol{Q}_i^\top$ so $\boldsymbol{Q}'_i \boldsymbol{a} = \boldsymbol{Q}'_i \boldsymbol{Q}_i \boldsymbol{P} \boldsymbol{a} = \boldsymbol{a}$. Due to the property of anchors (Lemma 4.2), we have $\boldsymbol{Q}'_i \boldsymbol{x}_i = \boldsymbol{x}_i$. Notice that $\boldsymbol{x}_i = \boldsymbol{Q}'_i \boldsymbol{x}_i = \boldsymbol{P}^\top \boldsymbol{Q}_i^\top \boldsymbol{Q}_i \boldsymbol{x}'_i = \boldsymbol{P}\boldsymbol{x}'_i$. Therefore, we can conclude the proof as we have found a permutation matrix $\boldsymbol{P}$ that simultaneously aligns $\boldsymbol{x}_i$ and $\boldsymbol{x}'_i$ for every $i \in [D]$, i.e., $\boldsymbol{X} = [\boldsymbol{x}_1 \quad \cdots \quad \boldsymbol{x}_D] = [\boldsymbol{P}\boldsymbol{x}_1 \quad \cdots \quad \boldsymbol{P}\boldsymbol{x}_D] = \boldsymbol{P}\boldsymbol{X}'$. $\square$

Next, we need to examine how many weights are needed to construct an anchor via linear combining all the existing channels. We restate Lemma 4.4 with more specifications as well as a mathematical device to prove it as below:

**Lemma C.3.** *Consider $D$ linearly independent weight vectors $\{\boldsymbol{w}_1, \cdots, \boldsymbol{w}_D \in \mathbb{R}^D\}$. Then for every $p, q \in [N]$ such that $\boldsymbol{x}^{(p)} \ne \boldsymbol{x}^{(q)}$, there exists $\boldsymbol{w}_j, j \in [D]$, such that $\boldsymbol{x}^{(p)\top} \boldsymbol{w}_j \ne \boldsymbol{x}^{(q)\top} \boldsymbol{w}_j$.*

*Proof.* This is the simple fact of full-rank linear systems. Prove by contradiction. Suppose for $\forall j \in [D]$ we have $\boldsymbol{x}^{(p)\top} \boldsymbol{w}_j = \boldsymbol{x}^{(q)\top} \boldsymbol{w}_j$. Then we form a linear system: $\boldsymbol{x}^{(p)\top} [\boldsymbol{w}_1 \quad \cdots \quad \boldsymbol{w}_D] = \boldsymbol{x}^{(q)\top} [\boldsymbol{w}_1 \quad \cdots \quad \boldsymbol{w}_D]$. Since $\boldsymbol{w}_1, \cdots, \boldsymbol{w}_D$ are linearly independent, it yields $\boldsymbol{x}^{(p)} = \boldsymbol{x}^{(q)}$, which reaches the contradiction. $\square$

**Lemma C.4** (Anchor Construction)**.** *Consider a set of weight vectors $\{\boldsymbol{w}_1, \cdots, \boldsymbol{w}_{K_1} \in \mathbb{R}^D\}$ with $K_1 = N(N-1)(D-1)/2 + 1$, of which every $D$-length subset, i.e., $\{\boldsymbol{w}_j : \forall j \in \mathcal{J}\}, \forall \mathcal{J} \subseteq [K_1], |\mathcal{J}| = D$, is linearly independent, then for every data matrix $\boldsymbol{X} \in \mathbb{R}^{N \times D}$, there exists $j^* \in [K_1]$, $\boldsymbol{X}\boldsymbol{w}_{j^*}$ is an anchor of $\boldsymbol{X}$.*

*Proof.* Define a set of pairs which an anchor needs to distinguish: $\mathcal{I} = \{(p, q) : \boldsymbol{x}^{(p)} \ne \boldsymbol{x}^{(q)}\} \subseteq [N]^2$ Consider a $D$-length subset $\mathcal{J} \subseteq [K]$ with $|\mathcal{J}| = D$. Since $\{\boldsymbol{w}_j : \forall j \in \mathcal{J}\}$ are linear independent, we assert by Lemma C.3 that for every pair $(p, q) \in \mathcal{I}$, there exists $j \in \mathcal{J}, \boldsymbol{x}^{(p)\top} \boldsymbol{w}_j \ne \boldsymbol{x}^{(q)\top} \boldsymbol{w}_j$. It is equivalent to claim: for every pair $(p, q) \in \mathcal{I}$, at most $D - 1$ many $\boldsymbol{w}_j, j \in [K_1]$ satisfy $\boldsymbol{x}^{(p)\top} \boldsymbol{w}_j = \boldsymbol{x}^{(q)\top} \boldsymbol{w}_j$. Based on pigeon-hold principle, as long as $K_1 \ge N(N-1)(D-1)/2 + 1 = (D-1)\binom{N}{2} + 1 \ge (D-1)|\mathcal{I}| + 1$, there must exist $j^* \in [K_1]$ such that $\boldsymbol{x}^{(p)\top} \boldsymbol{w}_{j^*} \ne \boldsymbol{x}^{(q)\top} \boldsymbol{w}_{j^*}$ for $\forall (p, q) \in \mathcal{I}$. By Definition 4.1, $\boldsymbol{X}\boldsymbol{w}_{j^*}$ generates an anchor. $\square$

**Proposition C.5.** *The linear independence condition in Lemma C.4 can be satisfied with probability one by drawing i.i.d. Gaussian random vectors $\boldsymbol{w}_1, \cdots, \boldsymbol{w}_{K_1} \overset{i.i.d.}{\sim} \mathcal{N}(\boldsymbol{0}, \boldsymbol{I})$.*

*Proof.* We first note that generating a $D \times K_1$ Gaussian random matrix ($D \le K_1$) is equivalent to drawing a matrix with respect to a probability measure defined over $\mathcal{M} = \{\boldsymbol{X} \in \mathbb{R}^{D \times K} : \text{rank}(\boldsymbol{X}) \le D\}$. Since rank-$D$ matrices are dense in $\mathcal{M}$ [53, 54], we can conclude

492 that for $\forall \mathcal{J} \subseteq [K_1], |\mathcal{J}| = D$, $\mathbb{P}(\{\boldsymbol{w}_j : j \in \mathcal{J}\}$ are linearly independent$) = 1$. By union bound,
493 $\mathbb{P}(\{\boldsymbol{w}_j : j \in \mathcal{J}\}$ for all $\mathcal{J} \in [K], |\mathcal{J}| = D$ are linearly independent$) = 1$. $\qquad\qquad\square$

494 We also restate Lemma 4.5 in the following lemma to demonstrate the weight construction for anchor
495 matching:

496 **Lemma C.6** (Anchor Matching). *Consider a group of coefficients* $\Gamma = \{\gamma_1, \cdots, \gamma_{K_2} \in \mathbb{R}\}$ *with*
497 $\gamma_i \neq 0, \forall i \in [K_2]$, $\gamma_i \neq \gamma_j, \forall i, j \in [K_2]$, *and* $K_2 = N(N-1) + 1$ *such that for all 4-tuples*
498 $(\gamma_i, \gamma_j, \gamma_k, \gamma_l) \subset \Gamma$, *if* $\gamma_i \neq \gamma_j, \gamma_i \neq \gamma_k$ *then* $\gamma_i/\gamma_j \neq \gamma_k/\gamma_l$. *It must hold that: Given any*
499 $\boldsymbol{x}, \boldsymbol{x}', \boldsymbol{y}, \boldsymbol{y}' \in \mathbb{R}^N$ *such that* $\boldsymbol{x} \sim \boldsymbol{x}'$ *and* $\boldsymbol{y} \sim \boldsymbol{y}'$, *if* $(\boldsymbol{x} - \gamma_k \boldsymbol{y}) \sim (\boldsymbol{x}' - \gamma_k \boldsymbol{y}')$ *for every* $k \in [K_2]$,
500 *then* $[\boldsymbol{x} \quad \boldsymbol{y}] \sim [\boldsymbol{x}' \quad \boldsymbol{y}']$.

501 *Proof.* We note that $\boldsymbol{x} \sim \boldsymbol{x}'$ and $\boldsymbol{y} \sim \boldsymbol{y}'$ imply that there exist $\boldsymbol{P}_x, \boldsymbol{P}_y \in \Pi(N)$ such that $\boldsymbol{P}_x \boldsymbol{x} = \boldsymbol{x}'$
502 and $\boldsymbol{P}_y \boldsymbol{y} = \boldsymbol{y}'$. Also $(\boldsymbol{x} - \gamma_k \boldsymbol{y}) \sim (\boldsymbol{x}' - \gamma_k \boldsymbol{y}'), \forall k \in [K_2]$ implies there exists $\boldsymbol{Q}_k \in \Pi(N), \forall k \in$
503 $[K_2]$ such that $\boldsymbol{Q}_k(\boldsymbol{x} - \gamma_k \boldsymbol{y}) = \boldsymbol{x}' - \gamma_k \boldsymbol{y}'$. Substituting the former to the latter, we can obtain:

$$\left(\boldsymbol{I} - \boldsymbol{Q}_k^\top \boldsymbol{P}_x\right) \boldsymbol{x} = \gamma_k \left(\boldsymbol{I} - \boldsymbol{Q}_k^\top \boldsymbol{P}_x\right) \boldsymbol{y}, \quad \forall k \in [K_2], \tag{15}$$

504 where for each $k \in [K_2]$, Eq. (15) corresponds to $N$ equalities as follows. Here, we let $(\boldsymbol{Z})_i$ denote
505 the $i$th column of the matrix $\boldsymbol{Z}$.

$$(\boldsymbol{I} - \boldsymbol{Q}_k^\top \boldsymbol{P}_x)_1^\top \boldsymbol{x} = \gamma_k (\boldsymbol{I} - \boldsymbol{Q}_k^\top \boldsymbol{P}_x)_1^\top \boldsymbol{y}$$
$$\vdots \tag{16}$$
$$(\boldsymbol{I} - \boldsymbol{Q}_k^\top \boldsymbol{P}_x)_N^\top \boldsymbol{x} = \gamma_k (\boldsymbol{I} - \boldsymbol{Q}_k^\top \boldsymbol{P}_x)_N^\top \boldsymbol{y}$$

506 We compare and entries in $\boldsymbol{x} = [\cdots x_p \cdots]^\top$ and for each entry index $p \in [N]$, we define a set of non-
507 zero pairwise differences between $x_p$ and other entries in $\boldsymbol{x}$: $\mathcal{D}_x^{(p)} = \{x_p - x_q : q \in [N], x_p \neq x_q\}$.
508 Similarly, for $\boldsymbol{y}$, we define $\mathcal{D}_y^{(p)} = \{y_p - y_q : q \in [N], y_p \neq y_q\}$. We note that for every $n \in [N]$,
509 either $(\boldsymbol{I} - \boldsymbol{Q}_k^\top \boldsymbol{P}_x)_n^\top \boldsymbol{x} = 0$ or $(\boldsymbol{I} - \boldsymbol{Q}_k^\top \boldsymbol{P}_x)_n^\top \boldsymbol{x} \in \mathcal{D}_x^{(p)}$ for some $p \in [N]$ as $(\boldsymbol{Q}_k^\top \boldsymbol{P}_x)_n^\top \boldsymbol{x}$ is some
510 $x_q$. Then it is sufficient to show there must exist $k \in [K_2]$ such that none of equations in Eq. (16) can
511 be induced by some elements in $\mathcal{D}_x^{(p)}$, i.e.,

$$\exists k^* \in [K_2], \forall p, n \in [N] \text{ such that } (\boldsymbol{I} - \boldsymbol{Q}_{k^*}^\top \boldsymbol{P}_x)_n^\top \boldsymbol{x} \notin \mathcal{D}_x^{(p)}. \tag{17}$$

512 This is because Eq. (17) implies:

$$(\boldsymbol{I} - \boldsymbol{Q}_{k^*} \boldsymbol{P}_x)^\top \boldsymbol{x} = \boldsymbol{0} \quad \Rightarrow \quad \boldsymbol{x} = \boldsymbol{Q}_{k^*}^\top \boldsymbol{P}_x \boldsymbol{x} = \boldsymbol{Q}_{k^*}^\top \boldsymbol{x}',$$
$$(\text{Since } \gamma_k \neq 0, \forall k \in [K_2]) \quad (\boldsymbol{I} - \boldsymbol{Q}_{k^*} \boldsymbol{P}_y)^\top \boldsymbol{y} = \boldsymbol{0} \quad \Rightarrow \quad \boldsymbol{y} = \boldsymbol{Q}_{k^*}^\top \boldsymbol{P}_y \boldsymbol{y} = \boldsymbol{Q}_{k^*}^\top \boldsymbol{y}',$$

513 which is $[\boldsymbol{x} \quad \boldsymbol{y}] = \boldsymbol{Q}_{k^*}^\top [\boldsymbol{x}' \quad \boldsymbol{y}']$.

514 To show Eq. (17), we construct $N$ bipartite graphs $\mathcal{G}^{(p)} = (\mathcal{D}_x^{(p)}, \mathcal{D}_y^{(p)}, \mathcal{E}^{(p)})$ for $p \in [N]$ where each
515 $\alpha \in \mathcal{D}_x^{(p)}$ or each $\beta \in \mathcal{D}_y^{(p)}$ is viewed as a node and $(\alpha, \beta) \in \mathcal{E}^{(p)}$ gives an edge if $\alpha = \gamma_k \beta$ for
516 some $k \in [K_2]$. Then we prove the existence of $k^*$ via seeing a contradiction that does counting the
517 number of connected pairs $(\alpha, \beta)$ from two perspectives.

518 **Perspective of** $\mathcal{D}_x^{(p)}$. We argue that for $\forall p \in [N]$ and arbitrary $\alpha_1, \alpha_2 \in \mathcal{D}_x^{(p)}, \alpha_1 \neq \alpha_2$, there
519 exists at most one $\beta \in \mathcal{D}_y^{(p)}$ such that $(\alpha_1, \beta) \in \mathcal{E}^{(p)}$ and $(\alpha_2, \beta) \in \mathcal{E}^{(p)}$. Otherwise, suppose there
520 exists $\beta' \in \mathcal{D}_y^{(p)}, \beta' \neq \beta$ such that $(\alpha_1, \beta') \in \mathcal{E}^{(p)}$ and $(\alpha_2, \beta') \in \mathcal{E}^{(p)}$. Then we have $\alpha_1 = \gamma_i \beta$,
521 $\alpha_2 = \gamma_j \beta$, $\alpha_1 = \gamma_k \beta'$, and $\alpha_2 = \gamma_l \beta'$ for some $\gamma_i, \gamma_j, \gamma_k, \gamma_l \in \Gamma$, which is $\gamma_i/\gamma_k = \gamma_k/\gamma_l$. As
522 $\alpha_1 \neq \alpha_2$, it is obvious that $\gamma_i \neq \gamma_j$. Similarly, we have $\gamma_i \neq \gamma_k$. Altogether, it contradicts our
523 assumption on $\Gamma$. Therefore, $|\mathcal{E}^{(p)}| \leq 2 \max\{|\mathcal{D}_x^{(p)}|, |\mathcal{D}_y^{(p)}|\} \leq 2(N-1)$. And the total edge number
524 of all bipartite graphs should be less than $2N(N-1)$.

**Perspective of $\Gamma$.** We note that if for some $k \in [K_2]$ that makes $(\boldsymbol{I} - \boldsymbol{Q}_k^\top \boldsymbol{P}_x)_n^\top \boldsymbol{x} \in \mathcal{D}_x^{(p)}$ for some $p, n \in [N]$, i.e., $(\boldsymbol{I} - \boldsymbol{Q}_k^\top \boldsymbol{P}_x)_n^\top \boldsymbol{x} = \gamma_k (\boldsymbol{I} - \boldsymbol{Q}_k^\top \boldsymbol{P}_y)_n^\top \boldsymbol{y} \neq 0$, this $\gamma_k$ contributes at least two edges in the entire bipartite graph, i.e., there being another $n' \in [N]$, $(\boldsymbol{I} - \boldsymbol{Q}_k^\top \boldsymbol{P}_x)_{n'}^\top \boldsymbol{x} = \gamma_k (\boldsymbol{I} - \boldsymbol{Q}_k^\top \boldsymbol{P}_y)_{n'}^\top \boldsymbol{y} \neq 0$. Otherwise, there exists a unique $n^* \in [N]$ such that $(\boldsymbol{I} - \boldsymbol{Q}_k^\top \boldsymbol{P}_x)_{n^*}^\top \boldsymbol{x} \in \mathcal{D}_x^{(p)}(\neq 0)$ and $(\boldsymbol{I} - \boldsymbol{Q}_k^\top \boldsymbol{P}_x)_n^\top \boldsymbol{x} = 0$ for all $n \neq n^*$. This cannot be true because $\boldsymbol{1}^\top (\boldsymbol{I} - \boldsymbol{Q}_k^\top \boldsymbol{P}_x) \boldsymbol{x} = 0$. By which, if $\forall k \in [K_2], \exists p, n \in [N]$ such that $(\boldsymbol{I} - \boldsymbol{Q}_k^\top \boldsymbol{P}_x)_n^\top \boldsymbol{x} \in \mathcal{D}_x^{(p)}$ (i.e., Eq. (17) is always false), then the total number of edges is at least $2K_2 = 2N(N-1) + 2$.

Hereby, we conclude the proof by the contradiction, in which the minimal count of edges $2K_2$ by **Perspective of $\Gamma$** already surpasses the maximal number $2N(N-1)$ by **Perspective of $\mathcal{D}_x^{(p)}$**. $\qquad\square$

*Remark* C.7. A handy choice of $\Gamma$ in Lemma C.6 are prime numbers, which are provably positive, infinitely many, and not divisible by each other.

We wrap off this section by formally stating and proving the injectivity statement of the LP embedding layer.

**Theorem C.8.** *Suppose $\phi : \mathbb{R}^D \to \mathbb{R}^L$ admits the form of Eq. (2) where $L = KN \leq N^5 D^2$, $K = D + K_1 + DK_1K_2$ and $\boldsymbol{W} = \begin{bmatrix} \boldsymbol{w}_1^{(1)} & \cdots & \boldsymbol{w}_D^{(1)} & \boldsymbol{w}_1^{(2)} & \cdots & \boldsymbol{w}_{K_1}^{(2)} & \cdots & \boldsymbol{w}_{i,j,k}^{(3)} & \cdots \end{bmatrix}$ is constructed as follows:*

*1. Let the first group of weights $\boldsymbol{w}_1^{(1)} = \boldsymbol{e}_1, \cdots, \boldsymbol{w}_D^{(1)} = \boldsymbol{e}_D$ to buffer the original features, where $\boldsymbol{e}_i$ is the $i$-th canonical basis.*

*2. Choose the second group of linear weights, $\boldsymbol{w}_1^{(2)}, \cdots, \boldsymbol{w}_{K_1}^{(2)}$ for $K_1$ as large as $N(N-1)(D-1)/2 + 1$, such that the conditions in Lemma C.4 are satisfied.*

*3. Design the third group of weights $\boldsymbol{w}_{i,j,k}^{(3)}$ for $i \in [D], j \in [K_1], k \in [K_2]$ where $\boldsymbol{w}_{i,j,k}^{(3)} = \boldsymbol{e}_i - \gamma_k \boldsymbol{w}_j^{(2)}$, $K_2 = N(N-1) + 1$, and $\gamma_k, k \in [K_2]$ are chosen such that conditions in Lemma C.6 are satisfied.*

*Then $\sum_{i=1}^N \phi(\boldsymbol{x}^{(i)})$ is injective (cf. Definition A.5).*

*Proof.* Suppose $\sum_{i=1}^N \phi(\boldsymbol{x}^{(i)}) = \sum_{i=1}^N \phi(\boldsymbol{x'}^{(i)})$ for some $\boldsymbol{X}, \boldsymbol{X'} \in \mathbb{R}^{N \times D}$. Due to the property of power mapping (cf. Lemma 2.8):

$$\sum_{i=1}^N \phi(\boldsymbol{x}^{(i)}) = \sum_{i=1}^N \phi(\boldsymbol{x'}^{(i)}) \Rightarrow \boldsymbol{X}\boldsymbol{w}_i^{(1)} \sim \boldsymbol{X'}\boldsymbol{w}_i^{(1)}, \forall i \in [D], \boldsymbol{X}\boldsymbol{w}_i^{(2)} \sim \boldsymbol{X'}\boldsymbol{w}_i^{(2)}, \forall i \in [K_1], \quad (18)$$

$$\boldsymbol{X}\boldsymbol{w}_{i,j,k}^{(3)} \sim \boldsymbol{X'}\boldsymbol{w}_{i,j,k}^{(3)}, \forall i \in [D], j \in [K_1], k \in [K_2].$$

By Lemma C.4, it is guaranteed that there exists $j^* \in [K_1]$ such that $\boldsymbol{X}\boldsymbol{w}_{j^*}^{(2)}$ is an anchor, and according to Eq. (18), we have $\boldsymbol{X}\boldsymbol{w}_{j^*}^{(2)} \sim \boldsymbol{X'}\boldsymbol{w}_{j^*}^{(2)}$. By Lemma C.6, we induce:

$$\boldsymbol{X}\boldsymbol{w}_i^{(1)} \sim \boldsymbol{X'}\boldsymbol{w}_i^{(1)}, \forall i \in [D], \boldsymbol{X}\boldsymbol{w}_{j^*}^{(2)} \sim \boldsymbol{X'}\boldsymbol{w}_{j^*}^{(2)}, \boldsymbol{X}\boldsymbol{w}_{i,j,k}^{(3)} \sim \boldsymbol{X'}\boldsymbol{w}_{i,j,k}^{(3)}, \forall i \in [D], j \in [K_1], k \in [K_2]$$

$$\Rightarrow \begin{bmatrix} \boldsymbol{X}\boldsymbol{w}_{j^*}^{(2)} & \boldsymbol{x}_i \end{bmatrix} \sim \begin{bmatrix} \boldsymbol{X'}\boldsymbol{w}_{j^*}^{(2)} & \boldsymbol{x'}_i \end{bmatrix}, \forall i \in [D]. \quad (19)$$

Since $\boldsymbol{X}\boldsymbol{w}_{j^*}^{(2)}$ is an anchor, by union alignment (Lemma 4.3), we have:

$$\begin{bmatrix} \boldsymbol{X}\boldsymbol{w}_{j^*}^{(2)} & \boldsymbol{x}_i \end{bmatrix} \sim \begin{bmatrix} \boldsymbol{X'}\boldsymbol{w}_{j^*}^{(2)} & \boldsymbol{x'}_i \end{bmatrix}, \forall i \in [D] \Rightarrow \boldsymbol{X} \sim \boldsymbol{X'}. \quad (20)$$

Here $K = D + K_1 + DK_1K_2 \leq N^4 D^2$, thus $L = KN \leq N^5 D^2$, which concludes the proof. $\qquad\square$

## C.2 Continuity

Next, we show that under the construction of Theorem C.8, the inverse of $\sum_{i=1}^N \phi(\boldsymbol{x}^{(i)})$ is continuous. The main idea is to check the three conditions provided in Lemma 4.10:

**Corollary C.9.** *Consider channel-wise high-dimensional sum-of-power* $\widehat{\Psi_N}(\boldsymbol{X})$ : $(\mathbb{R}^{N \times K}/ \sim, d_F) \to (\mathbb{R}^{NK}, d_\infty)$ *defined as below:*

$$\widehat{\Psi_N}(\boldsymbol{X}) = \begin{bmatrix} \Psi_N(\boldsymbol{x}_1)^\top & \cdots & \Psi_N(\boldsymbol{x}_K)^\top \end{bmatrix}^\top \in (\mathbb{R}^{NK}, d_\infty), \tag{21}$$

*and an associated mapping* $\widehat{\Phi_N} : (\mathbb{R}^{NK}, d_\infty) \to (\mathbb{R}^N/ \sim, d_F)^K$:

$$\widehat{\Phi_N}(\boldsymbol{Z}) = \begin{bmatrix} \Psi_N^{-1}(\boldsymbol{z}_1) & \cdots & \Psi_N^{-1}(\boldsymbol{z}_K) \end{bmatrix}, \tag{22}$$

*where* $\boldsymbol{Z} = \begin{bmatrix} \boldsymbol{z}_1^\top & \cdots & \boldsymbol{z}_K^\top \end{bmatrix}^\top$, $\boldsymbol{z}_i \in \mathbb{R}^N, \forall i \in [K]$. *We denote the induced product metric over* $(\mathbb{R}^N/ \sim, d_F)^K$ *as* $d_F^K : (\mathbb{R}^N/ \sim)^K \times (\mathbb{R}^N/ \sim)^K \to \mathbb{R}_{\geq 0}$:

$$d_F^K(\boldsymbol{Z}, \boldsymbol{Z}') = \max_{i \in [K]} d_F(\boldsymbol{z}_i, \boldsymbol{z}'_i). \tag{23}$$

*Then the mapping* $\widehat{\Phi_N}$ *maps any bounded set in* $(\mathbb{R}^{NK}, d_\infty)$ *to a bounded set in* $(\mathbb{R}^N/ \sim, d_F)^K$.

*Proof.* Proved by noting that if $d_\infty(\boldsymbol{z}_i, \boldsymbol{z}'_i) \leq C_1$ for some $\boldsymbol{z}_i, \boldsymbol{z}'_i \in (\mathbb{R}^N, d_\infty), \forall i \in [K]$ and a constant $C_1 \geq 0$, then $d_F(\Psi_N^{-1}(\boldsymbol{z}_i), \Psi_N^{-1}(\boldsymbol{z}'_i)) \leq C_2, \forall i \in [K]$ for some constant $C_2 \geq 0$ by Lemma B.5 and Remark B.6. Finally, we have:

$$d_F^K(\widehat{\Phi_N}(\boldsymbol{Z}), \widehat{\Phi_N}(\boldsymbol{Z}')) = \max_{i \in [K]} d_F(\Psi_N^{-1}(\boldsymbol{z}_i), \Psi_N^{-1}(\boldsymbol{z}'_i)) \leq C_2,$$

which is also bounded above. $\qquad\qquad\qquad\qquad\qquad\qquad\qquad\qquad\qquad\qquad\qquad\qquad\qquad$ $\square$

Now we are ready to present and prove the continuity of the LP embedding layer.

**Theorem C.10.** *Suppose $\phi$ admits the form of Eq.* (2) *and follows the construction in Theorem C.8, then the inverse of LP-induced sum-pooling* $\sum_{i=1}^N \phi(\boldsymbol{x}^{(i)})$ *is continuous.*

*Proof.* The proof is done by invoking Lemma 4.10. First of all, the inverse of $\Psi(\boldsymbol{X}) = \sum_{i=1}^N \phi(\boldsymbol{x}^{(i)})$, denoted as $\Psi^{-1} : (\mathbb{R}^{NK}, d_\infty) \to (\mathbb{R}^{N \times D}/ \sim, d_F)$, exists due to Theorem C.8. By Lemma A.4, any closed and bounded subset of $(\mathbb{R}^{N \times D}/ \sim, d_F)$ is compact. Trivially, $\Psi(\boldsymbol{X})$ is continuous. Then it remains to show the condition **(c)** in Lemma 4.10. We decompose $\Psi^{-1}$ into two mappings following the similar idea of proving its existence:

$$(\mathbb{R}^{NK}, d_\infty) \xrightarrow{\widehat{\Phi_N}} (\mathbb{R}^N/ \sim, d_F)^K \xrightarrow{\pi} (\mathbb{R}^{N \times D}/ \sim, d_F) ,$$

where $\widehat{\Phi_N}$ is defined in Eq. (22) and $\pi$ exists due to Eqs. (19) and (20) in Theorem C.8. Also according to our construction in Theorem C.8, the first identity block induces that: for any $\boldsymbol{Z} \in (\mathbb{R}^N/ \sim, d_F)^K$, $\boldsymbol{z}_i \sim \pi(\boldsymbol{Z})_i$ for every $i \in [D]$. Therefore, $\forall \boldsymbol{Z}, \boldsymbol{Z}' \in (\mathbb{R}^N/ \sim, d_F)^K$ such that $d_F^K(\boldsymbol{Z}, \boldsymbol{Z}') \leq C$ for some constant $C > 0$, we have:

$$d_F(\pi(\boldsymbol{Z}), \pi(\boldsymbol{Z}')) \leq \max_{i \in [D]} d_F(\boldsymbol{z}_i, \boldsymbol{z}'_i) \leq d_F^K(\boldsymbol{Z}, \boldsymbol{Z}') \leq C, \tag{24}$$

which implies $\pi$ maps every bounded set in $(\mathbb{R}^N/ \sim, d_F)^K$ to a bounded set in $(\mathbb{R}^{N \times D}/ \sim, d_F)$. Now we conclude the proof by the following chain of argument:

$$\mathcal{Z} \subseteq (\mathbb{R}^{NK}, d_\infty) \text{ is bounded} \xRightarrow{\text{Corollary C.9}} \widehat{\Phi_N}(\mathcal{Z}) \text{ is bounded} \xRightarrow{\text{Eq. (24)}} \pi \circ \widehat{\Phi_N}(\mathcal{Z}) \text{ is bounded.}$$

$\qquad\qquad\qquad\qquad\qquad\qquad\qquad\qquad\qquad\qquad\qquad\qquad\qquad\qquad\qquad\qquad\qquad\qquad\qquad\qquad$ $\square$

### C.3 Lower Bound for Injectivity

In this section, we prove Theorem 4.6 which shows that $K \geq D + 1$ is necessary for injectivity of LP-induced sum-pooling when $D \geq 2$. Our argument mainly generalizes Lemma 2 of [55] to our equivalence class. To proceed our argument, we define the linear subspace $\boldsymbol{V}$ by vectorizing $\begin{bmatrix} \boldsymbol{X}\boldsymbol{w}_1 & \cdots & \boldsymbol{X}\boldsymbol{w}_K \end{bmatrix}$ as below:

$$\mathcal{V} := \left\{ \begin{bmatrix} \boldsymbol{X}\boldsymbol{w}_1 \\ \vdots \\ \boldsymbol{X}\boldsymbol{w}_K \end{bmatrix} : \boldsymbol{X} \in \mathbb{R}^{N \times D} \right\} = \mathcal{R}\left( \begin{bmatrix} (\boldsymbol{w}_1 \otimes \boldsymbol{I}_N) \\ \vdots \\ (\boldsymbol{w}_K \otimes \boldsymbol{I}_N) \end{bmatrix} \right), \tag{25}$$

where $\mathcal{R}(\boldsymbol{Z})$ denotes the column space of $\boldsymbol{Z}$ and $\otimes$ is the Kronecker product. $\mathcal{V}$ is a linear subspace of $\mathbb{R}^{NK}$ with dimension at most $\mathbb{R}^{ND}$, characterized by $\boldsymbol{w}_1, \cdots, \boldsymbol{w}_K \in \mathbb{R}^D$. For the sake of notation simplicity, we denote $\Pi(N)^{\otimes K} = \{\text{diag}(\boldsymbol{Q}_1, \cdots, \boldsymbol{Q}_K) : \forall \boldsymbol{Q}_1, \cdots, \boldsymbol{Q}_K \in \Pi(N)\}$, and $\boldsymbol{I}_K \otimes \Pi(N) = \{\boldsymbol{I}_K \otimes \boldsymbol{Q} : \forall \boldsymbol{Q} \in \Pi(N)\}$. Next, we define the notion of unique recoverability [55] as below:

**Definition C.11** (Unique Recoverability). The subspace $\mathcal{V}$ is called uniquely recoverable under $\boldsymbol{Q} \in \Pi(N)^{\otimes K}$ if whenever $\boldsymbol{x}, \boldsymbol{x}' \in \mathcal{V}$ satisfy $\boldsymbol{Q}\boldsymbol{x} = \boldsymbol{x}'$, there exists $\boldsymbol{P} \in \boldsymbol{I}_K \otimes \Pi(N)$, $\boldsymbol{P}\boldsymbol{x} = \boldsymbol{x}'$.

Subsequently, we derive a necessary condition for the unqiue recoverability:

**Lemma C.12.** *A linear subspace $\mathcal{V} \subseteq \mathbb{R}^{NK}$ is uniquely recoverable under $\boldsymbol{Q} \in \Pi(N)^{\otimes K}$ only if there exists $\boldsymbol{P} \in \boldsymbol{I}_K \otimes \Pi(N)$, $\boldsymbol{Q}(\mathcal{V}) \cap \mathcal{V} \subset \mathcal{E}_{\boldsymbol{Q}\boldsymbol{P}^\top, \lambda=1}$, where $\mathcal{E}_{\boldsymbol{Q}\boldsymbol{P}^\top, \lambda}$ denotes the eigenspace corresponding to the eigenvalue $\lambda$.*

*Proof.* We first show that $\boldsymbol{Q}(\mathcal{V}) \cap \mathcal{V} \subseteq \bigcup_{\boldsymbol{P} \in \boldsymbol{I}_K \otimes \Pi(K)} \mathcal{E}_{\boldsymbol{Q}\boldsymbol{P}^\top, \lambda=1}$. Since the LHS is a subspace, while the RHS is a union of subspaces, there exists $\boldsymbol{P} \in \boldsymbol{I}_K \otimes \Pi(K)$ such that $\boldsymbol{Q}(\mathcal{V}) \cap \mathcal{V} \subseteq \mathcal{E}_{\boldsymbol{Q}\boldsymbol{P}^\top, \lambda=1}$. Then it remains to show $\boldsymbol{Q}(\mathcal{V}) \cap \mathcal{V} \subseteq \bigcup_{\boldsymbol{P} \in \boldsymbol{I}_K \otimes \Pi(K)} \mathcal{E}_{\boldsymbol{Q}\boldsymbol{P}^\top, \lambda=1}$.

Proved by contradiction. Suppose there exists $\boldsymbol{x} \in \boldsymbol{Q}(\mathcal{V}) \cap \mathcal{V}$ but $\boldsymbol{x} \notin \bigcup_{\boldsymbol{P} \in \boldsymbol{I}_K \otimes \Pi(K)} \mathcal{E}_{\boldsymbol{Q}\boldsymbol{P}^\top, \lambda=1}$. Or equivalently, there exists $\boldsymbol{x}' \in \mathcal{V}$ and $\boldsymbol{x} = \boldsymbol{Q}\boldsymbol{x}'$, while for $\forall \boldsymbol{P} \in \boldsymbol{I}_K \otimes \Pi(N)$, $\boldsymbol{x} \neq \boldsymbol{Q}\boldsymbol{P}^\top \boldsymbol{x}$. This implies $\boldsymbol{Q}^\top \boldsymbol{x} = \boldsymbol{x}' \neq \boldsymbol{P}\boldsymbol{x}$ for $\forall \boldsymbol{P} \in \boldsymbol{I}_K \otimes \Pi(N)$. However, this contradicts the fact that $\mathcal{V} \subseteq \mathbb{R}^{NK}$ is uniquely recoverable (cf. Definition C.11). $\square$

We also introduce a useful Lemma C.13 that gets rid of the discussion on $\boldsymbol{Q}$ in the inclusion:

**Lemma C.13.** *Suppose $\mathcal{V} \subseteq \mathbb{R}^N$ is a linear subspace, and $\boldsymbol{A}$ is a linear mapping. $\boldsymbol{A}(\mathcal{V}) \cap \mathcal{V} \cap \mathcal{E}_{\boldsymbol{A}, \lambda} = \boldsymbol{0}$ if and only if $\mathcal{V} \cap \mathcal{E}_{\boldsymbol{A}, \lambda} = \boldsymbol{0}$.*

*Proof.* The sufficiency is straightforward. The necessity is shown by contradiction: Suppose $\mathcal{V} \cap \mathcal{E}_{\boldsymbol{A}, \lambda} \neq \boldsymbol{0}$, then there exists $\boldsymbol{x} \in \mathcal{V} \cap \mathcal{E}_{\boldsymbol{A}, \lambda}$ such that $\boldsymbol{x} \neq \boldsymbol{0}$. Then $\boldsymbol{A}\boldsymbol{x} = \lambda\boldsymbol{x}$ implies $\boldsymbol{x} \in \boldsymbol{A}(\mathcal{V})$. Hence, $\boldsymbol{x} \in \boldsymbol{A}(\mathcal{V}) \cap \mathcal{V} \cap \mathcal{E}_{\boldsymbol{A}, \lambda}$ which reaches the contradiction. $\square$

Now we are ready to present the proof of Theorem 4.6:

*Proof of Theorem 4.6.* Proved by contrapositive. First notice that, $\forall \boldsymbol{X}, \boldsymbol{X}' \in \mathbb{R}^{N \times D}, \boldsymbol{X}\boldsymbol{w}_i \sim \boldsymbol{X}'\boldsymbol{w}_i, \forall i \in [K] \Rightarrow \boldsymbol{X} \sim \boldsymbol{X}'$ holds if and only if $\dim \mathcal{V} = ND$ and $\mathcal{V}$ is uniquely recoverable under all possible $\boldsymbol{Q} \in \Pi(N)^{\otimes K}$. By Lemma C.12, for every $\boldsymbol{Q} \in \Pi(N)^{\otimes K}$, there exists $\boldsymbol{P} \in \boldsymbol{I}_K \otimes \Pi(N)$ such that $\boldsymbol{Q}(\mathcal{V}) \cap \mathcal{V} \subset \mathcal{E}_{\boldsymbol{Q}\boldsymbol{P}^\top, \lambda=1}$. This is $\boldsymbol{Q}(\mathcal{V}) \cap \mathcal{V} \cap \mathcal{E}_{\boldsymbol{Q}\boldsymbol{P}^\top, \lambda} = 0$ for all $\lambda \neq 1$. By Lemma C.13, we have $\mathcal{V} \cap \mathcal{E}_{\boldsymbol{Q}\boldsymbol{P}^\top, \lambda} = 0$ for all $\lambda \neq 1$. Then proof is concluded by discussing the dimension of ambient space $\mathbb{R}^{NK}$ such that an $ND$-dimensional subspace $\mathcal{V}$ can reside. To ensure $\mathcal{V} \cap \mathcal{E}_{\boldsymbol{Q}\boldsymbol{P}^\top, \lambda} = 0$ for all $\lambda \neq 1$, it is necessary that $\dim \mathcal{V} \leq \min_{\lambda \neq 1} \text{codim} \, \mathcal{E}_{\boldsymbol{Q}\boldsymbol{P}^\top, \lambda}$ for every $\boldsymbol{Q} \in \Pi(N)^{\otimes K}$ and its associated $\boldsymbol{P} \in \boldsymbol{I}_K \otimes \Pi(N)$. Relaxing the dependence between $\boldsymbol{Q}$ and $\boldsymbol{P}$, we derive the inequality:

$$ND = \dim \mathcal{V} \leq \min_{\boldsymbol{Q} \in \Pi(N)^{\otimes K}} \max_{\boldsymbol{P} \in \boldsymbol{I}_K \otimes \Pi(N)} \min_{\lambda \neq 1} \text{codim} \, \mathcal{E}_{\boldsymbol{Q}\boldsymbol{P}^\top, \lambda} \leq NK - 1, \tag{26}$$

where the last inequality considers the scenario where every non-one eigenspace is one-dimensional, which is achievable when $K \geq 2$. Hence, we can bound $K \geq D + 1/K$, i.e., $K \geq D + 1$. $\square$

# D    Proofs for LLE Embedding Layer

In this section, we present the complete proof for the LLE embedding layer (Eq. (3)). Similar to the LP embedding layer, we construct an LLE whose induced sum-pooling is injective following arguments in Sec. 4.2 and has continuous inverse with the techniques introduced in Sec. 4.3.

## D.1 Upper Bound for Injectivity

To prove the upper bound, we construct an LLE embedding layer with $L \leq 2N^2D^2$ output neurons such that its induced sum-pooling is injective. The main proof technique is to bind every pair of channel with complex numbers and invoke the injectiviy of sum-of-power mapping over the complex domain.

With Lemma B.2, we can prove Lemma 4.8 as below:

*Proof of Lemma 4.8.* If for any pair of vectors $\boldsymbol{x}_1, \boldsymbol{x}_2 \in \mathbb{R}^N, \boldsymbol{y}_1, \boldsymbol{y}_2 \in \mathbb{R}^N$ such that $\sum_{i \in [N]} \boldsymbol{x}_{1,i}^{l-k} \boldsymbol{x}_{2,i}^{k} = \sum_{i \in [N]} \boldsymbol{y}_{1,i}^{l-k} \boldsymbol{y}_{2,i}^{k}$ for every $l, k \in [N], 0 \leq k \leq l$, then for $\forall l \in [N]$,

$$\sum_{i=1}^{N} (\boldsymbol{x}_{1,i} + \boldsymbol{x}_{2,i}\sqrt{-1})^l = \sum_{i=1}^{N} \sum_{k=0}^{l} (\sqrt{-1})^k \boldsymbol{x}_{1,i}^{l-k} \boldsymbol{x}_{2,i}^{k} = \sum_{k=0}^{l} (\sqrt{-1})^k \left( \sum_{i=1}^{N} \boldsymbol{x}_{1,i}^{l-k} \boldsymbol{x}_{2,i}^{k} \right) \qquad (27)$$

$$= \sum_{k=0}^{l} (\sqrt{-1})^k \left( \sum_{i=1}^{N} \boldsymbol{y}_{1,i}^{l-k} \boldsymbol{y}_{2,i}^{k} \right)$$

$$= \sum_{i=1}^{N} (\boldsymbol{y}_{1,i} + \boldsymbol{y}_{2,i}\sqrt{-1})^l$$

$$= \psi_N(\boldsymbol{y}_1 + \boldsymbol{y}_2\sqrt{-1})$$

Then by Lemma B.2, we have $(\boldsymbol{x}_1 + \boldsymbol{x}_2\sqrt{-1}) \sim (\boldsymbol{y}_1 + \boldsymbol{y}_2\sqrt{-1})$, which is essentially $[\boldsymbol{x}_1 \quad \boldsymbol{x}_2] \sim [\boldsymbol{y}_1 \quad \boldsymbol{y}_2]$. □

Now we are ready to prove the injectiviy of the LLE layer.

**Theorem D.1.** *Suppose* $\phi : \mathbb{R}^D \to \mathbb{R}^L$ *admits the form of Eq. (3) and* $\boldsymbol{W} = [\cdots \quad \boldsymbol{w}_{i,j,p,q} \quad \cdots] \in \mathbb{R}^{D \times L}, i \in [D], j \in [D], p \in [N], q \in [p+1]$ *is constructed as follows:*

$$\boldsymbol{w}_{i,j,p,q} = (q-1)\boldsymbol{e}_i + (p-q+1)\boldsymbol{e}_j, \qquad (28)$$

*where* $\boldsymbol{e}_i$ *is the $i$-th canonical basis. Then* $\sum_{i=1}^{N} \phi(\boldsymbol{x}^{(i)})$ *is injective (cf. Definition A.5).*

*Proof.* First of all, notice that $L = D^2 \sum_{p=1}^{N} (p+1) = D^2(N+3)N/2 \leq 2N^2D^2$. According to Eq. 8, we can rewrite for $\forall i \in [D], j \in [D], p \in [N], q \in [p+1]$

$$\phi(\boldsymbol{x})_{i,j,p,q} = \exp(\boldsymbol{w}_{i,j,p,q}^{\top} \log(\boldsymbol{x})) = \prod_{k=1}^{D} \boldsymbol{x}_k^{\boldsymbol{w}_{i,j,p,q,k}} = \boldsymbol{x}_i^{q-1} \boldsymbol{x}_j^{p-q+1}, \qquad (29)$$

Then for $\boldsymbol{X}, \boldsymbol{X}' \in \mathbb{R}^{N \times D}$, $\sum_{i \in [N]} \phi(\boldsymbol{x}^{(i)}) = \sum_{i \in [N]} \phi(\boldsymbol{x}'^{(i)})$ implies $\sum_{i \in [N]} \boldsymbol{x}_i^{q-1} \boldsymbol{x}_j^{p-q+1} = \sum_{i \in [N]} \boldsymbol{x}'_i^{q-1} \boldsymbol{x}'_j^{p-q+1}$ for $\forall i \in [D], j \in [D], p \in [N], q \in [p+1]$. By Lemma 4.8, we have $[\boldsymbol{x}_i \quad \boldsymbol{x}_j] \sim [\boldsymbol{x}'_i \quad \boldsymbol{x}'_j]$ for $\forall i \in [D], j \in [D]$. Finally, Lemma 4.9 directly yields $\boldsymbol{X} \sim \boldsymbol{X}'$. □

## D.2 Continuity

The proof idea of continuity for LLE layer shares the same outline with the LP layer.

**Corollary D.2.** *Consider the mapping* $\widetilde{\Phi_N} = \psi_N^{-1} \circ \tau : (\mathbb{R}^{N(N+3)/2}, d_\infty) \to (\mathbb{C}^N / \sim, d_F)$, *where* $\tau : (\mathbb{R}^{N(N+3)/2}, d_\infty) \to (\mathbb{C}^N, d_\infty)$ *is a linear mapping that combines sum of monomials to polynomials following Eq. (27). Construct the mapping* $\widehat{\Phi_N} : (\mathbb{R}^L, d_\infty) \to (\mathbb{C}^N / \sim, d_F)^{D^2}$ *based on* $\widetilde{\Phi_N}$:

$$\widehat{\Phi_N}(\boldsymbol{Z}) = \left[ \widetilde{\Phi_N}(\boldsymbol{z}_1) \quad \cdots \quad \widetilde{\Phi_N}(\boldsymbol{z}_{D^2}) \right], \qquad (30)$$

*where* $\boldsymbol{Z} = \left[ \boldsymbol{z}_1^{\top} \quad \cdots \quad \boldsymbol{z}_{D^2}^{\top} \right]^{\top} \in \mathbb{R}^L, \boldsymbol{z}_i \in \mathbb{R}^{(N(N+3)/2)}, \forall i \in [D^2]$. *We denote the induced product metric over* $(\mathbb{C}^N / \sim, d_F)^{D^2}$ *as* $d_F^{D^2} : (\mathbb{C}^N / \sim)^{D^2} \times (\mathbb{C}^N / \sim)^{D^2} \to \mathbb{R}_{\geq 0}$:

$$d_F^{D^2}(\boldsymbol{Z}, \boldsymbol{Z}') = \max_{i \in [D^2]} d_F(\boldsymbol{z}_i, \boldsymbol{z}'_i). \qquad (31)$$

*Then the mapping* $\widehat{\Phi_N}$ *maps any bounded set in* $(\mathbb{R}^L, d_\infty)$ *to a bounded set in* $(\mathbb{C}^N / \sim, d_F)^{D^2}$.

*Proof.* Since $\tau$ is a linear mapping, any $\boldsymbol{Z}, \boldsymbol{Z}' \in \mathbb{R}^L$ such that $d_\infty(\boldsymbol{z}_i, \boldsymbol{z}'_i) \leq C_1, \forall i \in [D^2]$ for some constant $C_1 \geq 0$, then $d_F(\tau(\boldsymbol{z}_i), \tau(\boldsymbol{z}'_i)) \leq C_2$ for some constant $C_2 \geq 0$. By Lemma B.3, $\psi_N^{-1}$ maps any bounded set in $(\mathbb{C}^N, d_\infty)$ to a bounded set in $(\mathbb{C}^N/\sim, d_F)$. This is $d_F(\widetilde{\Phi_N}(\boldsymbol{z}_i), \widetilde{\Phi_N}(\boldsymbol{z}'_i)) \leq C_2, \forall i \in [D^2]$ for some constant $C_2 \geq 0$. Finally, we have:

$$d_F^{D^2}(\widehat{\Phi_N}(\boldsymbol{Z}), \widehat{\Phi_N}(\boldsymbol{Z}')) = \max_{i \in [D^2]} d_F(\widetilde{\Phi_N}(\boldsymbol{z}_{i,j}), \widetilde{\Phi_N}(\boldsymbol{z}'_{i,j})) \leq C_2,$$

which is also bounded above. $\qquad\square$

**Theorem D.3.** *Suppose $\phi$ admits the form of Eq. (3) and follows the construction in Theorem D.1, then the inverse of LLE-induced sum-pooling $\sum_{i=1}^N \phi(\boldsymbol{x}^{(i)})$ is continuous.*

*Proof.* It is sufficient to verify three conditions in Lemma 4.10. First of all, we denote the inverse of $\Psi(\boldsymbol{X}) = \sum_{i=1}^N \phi(\boldsymbol{x}^{(i)})$, denoted as $\Psi^{-1}: (\mathbb{R}^L, d_\infty) \to (\mathbb{R}^{N \times D}/\sim, d_F)$, which exists thanks to Theorem D.1. By Lemma A.4, any closed and bounded subset of $(\mathbb{R}^{N \times D}/\sim, d_F)$ is compact. Obviously, $\Psi(\boldsymbol{X})$ is continuous. Then it remains to show the condition **(c)** in Lemma 4.10. Similar to Theorem C.10, we decompose $\Psi^{-1}$ into two mappings following the clue of proving its existence:

$$(\mathbb{R}^{NK}, d_\infty) \xrightarrow{\widehat{\Phi_N}} (\mathbb{C}^N/\sim, d_F)^{D^2} \xrightarrow{\pi} (\mathbb{R}^{N \times D}/\sim, d_F) \ ,$$

where $\widehat{\Phi_N}$ is defined in Eq. (30) and $\pi$ exists due to Theorem D.1. Also according to our construction in Theorem D.1, for any $\boldsymbol{Z} \subset (\mathbb{C}^N/\sim, d_F)^{D^2}$ and $\forall i, j \in [D]$, there exists $k \in [D^2]$ such that $(\pi(\boldsymbol{z})_i + \pi(\boldsymbol{z})_j \sqrt{-1}) \sim \boldsymbol{z}_k$. Therefore, $\forall \boldsymbol{Z}, \boldsymbol{Z}' \in (\mathbb{C}^N/\sim, d_F)^{D^2}$ such that $d_F^{D^2}(\boldsymbol{Z}, \boldsymbol{Z}') \leq C$ for some constant $C > 0$, we have:

$$d_F(\pi(\boldsymbol{Z}), \pi(\boldsymbol{Z}')) \leq \max_{i \in [D^2]} d_F(\boldsymbol{z}_i, \boldsymbol{z}'_i) \leq d_F^{D^2}(\boldsymbol{Z}, \boldsymbol{Z}') \leq C, \tag{32}$$

which implies $\pi$ maps every bounded set in $(\mathbb{C}^N/\sim, d_F)^K$ to a bounded set in $(\mathbb{R}^{N \times D}/\sim, d_F)$. Now we conclude the proof by the following chain of argument:

$$\mathcal{Z} \subseteq (\mathbb{R}^{NK}, d_\infty) \text{ is bounded} \xRightarrow{\text{Corollary D.2}} \widehat{\Phi_N}(\mathcal{Z}) \text{ is bounded} \xRightarrow{\text{Eq. (32)}} \pi \circ \widehat{\Phi_N}(\mathcal{Z}) \text{ is bounded.}$$

$\qquad\square$

# E  Extension to Permutation Equivariance

In this section, we prove Theorem 5.1, the extension of Theorem 3.1 to equivariant functions, following the similar arguments with [7]:

**Lemma E.1** ( [7,34]). *$f: \mathbb{R}^{N \times D} \to \mathbb{R}^N$ is a permutation-equivariant function if and only if there is a function $\rho: \mathbb{R}^{N \times D} \to \mathbb{R}$ that is permutation invariant to the last $N-1$ entries, such that $f(\boldsymbol{Z})_i = \rho(\boldsymbol{z}^{(i)}, \underbrace{\boldsymbol{z}^{(i+1)}, \cdots, \boldsymbol{z}^{(N)}, \cdots, \boldsymbol{z}^{(i-1)}}_{N-1}) \text{ for any } i \in [N].$*

*Proof.* (Sufficiency) Define $\pi: [N] \to [N]$ be an index mapping associated with the permutation matrix $\boldsymbol{P} \in \Pi(N)$ such that $\boldsymbol{PZ} = \left[\boldsymbol{z}^{(\pi(1))}, \cdots, \boldsymbol{z}^{(\pi(N))}\right]^\top$. Then we have:

$$f\left(\boldsymbol{z}^{(\pi(1))}, \cdots, \boldsymbol{z}^{(\pi(N))}\right)_i = \rho\left(\boldsymbol{z}^{(\pi(i))}, \boldsymbol{z}^{(\pi(i+1))}, \cdots, \boldsymbol{z}^{(\pi(N))}, \cdots, \boldsymbol{z}^{(\pi(i-1))}\right).$$

Since $\rho(\cdot)$ is invariant to the last $N-1$ entries, it can shown that:

$$f(\boldsymbol{PZ})_i = \rho\left(\boldsymbol{z}^{(\pi(i))}, \boldsymbol{z}^{(\pi(i+1))}, \cdots, \boldsymbol{z}^{(\pi(N))}, \cdots, \boldsymbol{z}^{(\pi(i-1))}\right) = f(\boldsymbol{Z})_{\pi(i)}.$$

(Necessity) Given a permutation-equivariant function $f: \mathbb{R}^{N \times D} \to \mathbb{R}^N$, we first expand it to the following form: $f(\boldsymbol{Z})_i = \rho_i(\boldsymbol{z}^{(1)}, \cdots, \boldsymbol{z}^{(N)})$. Permutation-equivariance means

685   $\rho_{\pi(i)}(\boldsymbol{z}^{(1)}, \cdots, \boldsymbol{z}^{(N)}) = \rho_i(\boldsymbol{z}^{\pi(1)}, \cdots, \boldsymbol{z}^{\pi(N)})$ for any permutation mapping $\pi$. Suppose given
686   an index $i \in [N]$, consider any permutation $\pi : [N] \to [N]$ such that $\pi(i) = i$. Then, we have:

$$\rho_i\left(\boldsymbol{z}^{(1)}, \cdots, \boldsymbol{z}^{(i)}, \cdots, \boldsymbol{z}^{(N)}\right) = \rho_{\pi(i)}\left(\boldsymbol{z}^{(1)}, \cdots, \boldsymbol{z}^{(i)}, \cdots, \boldsymbol{z}^{(N)}\right) = \rho_i\left(\boldsymbol{z}^{(\pi(1))}, \cdots, \boldsymbol{z}_i, \cdots, \boldsymbol{z}^{(\pi(N))}\right),$$

687   which implies $\rho_i : \mathbb{R}^{N \times D} \to \mathbb{R}$ must be invariant to the $N - 1$ elements other than the $i$-th element.
688   Now, consider a permutation $\pi$ where $\pi(1) = i$. Then we have:

$$\rho_i\left(\boldsymbol{z}^{(1)}, \boldsymbol{z}^{(2)}, \cdots, \boldsymbol{z}^{(N)}\right) = \rho_{\pi(1)}\left(\boldsymbol{z}^{(1)}, \boldsymbol{z}^{(2)}, \cdots, \boldsymbol{z}^{(N)}\right) = \rho_1\left(\boldsymbol{z}^{(\pi(1))}, \boldsymbol{z}^{(\pi(2))}, \cdots, \boldsymbol{z}^{(\pi(N))}\right)$$
$$= \rho_1\left(\boldsymbol{z}^{(i)}, \boldsymbol{z}^{(i+1)}, \cdots, \boldsymbol{z}^{(N)}, \cdots, \boldsymbol{z}^{(i-1)}\right),$$

689   where the last equality is due to the invariance to $N - 1$ elements, stated beforehand. This implies
690   two results. First, for all $i$, $\rho_i(\boldsymbol{z}^{(1)}, \boldsymbol{z}^{(2)}, \cdots, \boldsymbol{z}^{(i)}, \cdots, \boldsymbol{z}^{(N)}), \forall i \in [N]$ should be written in terms
691   of $\rho_1(\boldsymbol{z}^{(i)}, \boldsymbol{z}^{(i+1)}, \cdots, \boldsymbol{z}^{(N)}, \cdots, \boldsymbol{z}^{(i-1)})$. Moreover, $\rho_1$ is permutation invariant to its last $N - 1$
692   entries. Therefore, we just need to set $\rho = \rho_1$ and broadcast it accordingly to all entries. We conclude
693   the proof. $\qquad\qquad\square$

694   *Proof of Theorem 5.1 [7].* Sufficiency can be shown by verifying the equivariance. We conclude
695   the proof by showing the necessity with Lemma E.1. First we rewrite any permutation equivariant
696   function $f(\boldsymbol{x}^{(1)}, \cdots, \boldsymbol{x}^{(N)}) : \mathbb{R}^{N \times D} \to \mathbb{R}^N$ as:

$$f\left(\boldsymbol{x}^{(1)}, \cdots, \boldsymbol{x}^{(N)}\right)_i = \tau\left(\boldsymbol{x}^{(i)}, \boldsymbol{x}^{(i+1)}, \cdots, \boldsymbol{x}^{(N)}, \cdots, \boldsymbol{x}^{(i-1)}\right), \qquad (33)$$

697   where $\pi$ is invariant to the lask $N - 1$ elements, according to Lemma E.1. Given $\phi$ with either LP or
698   LLE architectures, $\Psi(\boldsymbol{X}) = \sum_{i=1}^N \phi(\boldsymbol{x}^{(i)})$ is injective and has continuous inverse if:

699   • $L \in [N(D+1), N^5 D^2]$ when $\phi$ admits LP architecture. (By Theorem C.8 and C.10).

700   • $L \in [ND, 2N^2 D^2]$ when $\phi$ admits LLE architecture. (By Theorem D.1 and D.3).

701   The proof proceeds by letting $\rho : \mathbb{R}^D \times \mathbb{R}^L \to \mathbb{R}$ take the form $\rho(\boldsymbol{x}, \boldsymbol{z}) = \tau(\boldsymbol{x}, \Phi^{-1}(\boldsymbol{z} - \phi(\boldsymbol{x})))$,
702   and observe that:

$$\tau\left(\boldsymbol{x}^{(i)}, \boldsymbol{x}^{(i+1)}, \cdots, \boldsymbol{x}^{(N)}, \cdots, \boldsymbol{x}^{(i-1)}\right) = \tau\left(\boldsymbol{x}^{(i)}, \Phi^{-1} \circ \Phi(\boldsymbol{x}^{(i+1)}, \cdots, \boldsymbol{x}^{(N)}, \cdots, \boldsymbol{x}^{(i-1)})\right)$$
$$= \tau\left(\boldsymbol{x}^{(i)}, \Phi^{-1}\left(\Phi\left(\boldsymbol{x}^{(1)}, \cdots, \boldsymbol{x}^{(N)}\right) - \phi(\boldsymbol{x}^{(i)})\right)\right)$$
$$= \rho\left(\boldsymbol{x}^{(i)}, \sum_{i=1}^N \phi(\boldsymbol{x}^{(i)})\right)$$

703   $\qquad\qquad\square$

# F   Extension to Complex Numbers

705   In this section, we formally introduce the nature extension of our Theorem 3.1 to the complex
706   numbers:

707   **Corollary F.1** (Extension to Complex Domain). *For any permutation-invariant function $f$ :*
708   $\mathcal{K}^{N \times D} \to \mathbb{R}$, $\mathcal{K} \subseteq \mathbb{C}$, *there exists continuous functions $\phi : \mathbb{C}^D \to \mathbb{R}^L$ and $\rho : \mathbb{R}^L \to \mathbb{C}$ such*
709   *that $f(\boldsymbol{X}) = \rho\left(\sum_{i \in [N]} \phi(\boldsymbol{x}^{(i)})\right)$ for every $j \in [N]$, where $L \in [2N(D+1), 4N^5 D^2]$ when $\phi$*
710   *admits LP architecture, and $L \in [2ND, 8N^2 D^2]$ when $\phi$ admits LLE architecture ($\mathcal{K} \in \mathbb{C}_{>0}$).*

711   *Proof.* We let $\phi$ first map complex features $\boldsymbol{x}^{(i)} \in \mathbb{C}^D, \forall i \in [N]$ to real features $\widetilde{\boldsymbol{x}}^{(i)} =$
712   $\left[\Re(\boldsymbol{x}^{(i)})^\top \quad \Im(\boldsymbol{x}^{(i)})^\top\right] \in \mathbb{R}^{2D}, \forall i \in [N]$ by divide the real and imaginary parts into separate
713   channels, then utilize either LP or LLE embedding layer to map $\widetilde{\boldsymbol{x}}^{(i)}$ to the latent space. The upper
714   bounds of desired latent space dimension are scaled by $4$ for both architectures due to the quadratic
715   dependence on $D$. Then the same proof of Theorems C.8, C.10, D.1, and D.3 applies. $\qquad\square$

## G  Connection to Unlabled Sensing

Unlabeled sensing [56], also known as linear regression without correspondence [55, 57–59], solves the linear system $\boldsymbol{y} = \boldsymbol{P}\boldsymbol{A}\boldsymbol{x}$ with a given measurement matrix $\boldsymbol{A} \in \mathbb{R}^{M \times N}$ and an unknown permutation $\boldsymbol{P} \in \Pi(M)$. [55, 56] show that as long as $\boldsymbol{A}$ is over-determinant ($M \geq 2N$), such problem is well-posed (i.e., has a unique solution) for almost all cases. Unlabeled sensing shares the similar structure with our LP embedding layer in which a linear layer lifts the feature space to a higher-dimensional ambient space, ensuring the solvability of alignment across each channel. However, our invertibility is defined between the set and embedding spaces, which differs from exact recovery of unknown variables desired in unlabeled sensing [55]. In fact, the well-posedness of unlabeled PCA [54], studying matrix completion with shuffle perturbations, shares the identical definition with our injectivity. But it is noteworthy that the results in [54] are only drawn over *a dense subset of the input space*, while ours are stronger in considering *all possible inputs*. Hence, our theory could potentially bring new insights into the field of unlabeled sensing, which may be of an independent interest.