# OpenReview forum: "Polynomial Width is Sufficient for Set Representation with High-dimensional Features"
_NeurIPS.cc/2023/Conference — Submitted to NeurIPS 2023_

### Official Review · Reviewer_S2JM · 2023-06-27

**Soundness:** 4 excellent
**Presentation:** 4 excellent
**Contribution:** 3 good
**Rating:** 6
**Confidence:** 4

**Summary:**

Summary: The paper proves that for symmetric neural networks, specifically DeepSets, there exist exact representations for symmetric functions, where the symmetric embedding layer width can be chosen to be polynomial in the set size and input dimension, rather than exponential as shown in stricter settings.

**Strengths:**

Strengths: The proof technique is clever, managing to continuously invert a specific symmetric embedding.  It’s also quite surprising, as the set of multisymmetric polynomial generators is exponentially large in N and D, which intuitively suggests that a map from the symmetric set to a strict subset of the generators wouldn’t be invertible.  The insight that this map can be inverted (at the price of perhaps being quite non-smooth) is a novel one.  I think this is a useful result in further understanding the capabilities of the DeepSets architecture specifically.


**Weaknesses:**

Weaknesses: I think the paper would benefit from a more robust discussion of the tradeoffs of this parameterization.  In particular, in the complex setting and under some standard network assumptions, the parameterization doesn’t have exponentially large width in the symmetric embedding layer L, but it must pay exponentially large width somewhere else.

For example, consider input dimension D = 1 and set size N > 1.  Assume N is odd, and let z denote a N-th principle root of unity.  Consider input sets x = 1/2 * (z^0, z^1, … z^{N-1}), and y = -x.  One can confirm that all the power sums p_k(x) = p_k(y) = 0 for 1 <= k <= N-1, and p_N(x) = N * (1/2)^N and p_N(y) = -N * (1/2)^N.

If psi_N is the map of the first N powersums as in definition 2.6, then we have d(psi_N(x), psi_N(y)) = N * (1/2)^{N-1}, but d(x,y) = O(1/sqrt(N)) (where this is using the appropriate notion of distance on sets, i.e. infinity norm modulo permutation).

All this to say, the inverse map psi_N^{-1} has a Lipschitz constant that is exponentially large in N.  So for a neural network of constant depth, with an activation with bounded Lipschitz constant and polynomially bounded weights, representing this function would require exponentially large width.

Of course this is a toy example.  But it’s extremely difficult to tell how non-smooth and nasty the parameterization given will be in general, especially when D > 1.  I understand the authors leave this question to future work, but in that case I think it’s useful to discuss that there’s no free lunch, and the given argument does not guarantee an efficient network overall.

More broadly, the parameterization is somewhat unrealistic.  The parameterization focuses specifically on mapping the set data into a symmetric embedding, inverting the embedding, and then feeding the original set information through another parameterized function.  So one struggles to see why to use DeepSets in the first place.  You still need to parameterize a symmetric function somehow - it would be silly to parameterize rho itself with DeepSets, but then how would you parameterize it?  The fact that, in practice, DeepSets works better than networks that don’t enforce the symmetry constraint suggests this parameterization is not a very practical model.


**Questions:**

N/A

---

> ### Author Rebuttal · Authors · 2023-08-10
>
> We sincerely thank reviewer S2JM for appreciating our proof technique and its deeper implication. We are also grateful for constructive suggestions to improve this manuscript. Please see our responses as below:
>
> **1. A more robust discussion of the tradeoffs of this parameterization.**
>
> We agree with reviewer's observation that given current construction, the inverse map $\Psi_N^{-1}$ may have an unbounded Lipschitz constant, which causes the complexity of $\rho$ may still grow exponentially. It is likely that the current construction still cannot bound the whole network size with polynomial many neurons for free, and we will add the discussion on this trade-off in our revision.
>
> **2. Parameterization is somewhat unrealistic.**
>
> We really appreciate the insights provided by the reviewer. We partially agree with the reviewer's argument but not all of them. First, this work is mainly to address an important theoretical question in this field. As the reviewer may also agree, the constructive proof may provides insights into not only just a particular architecture but also a broader idea of building homomorphism mappings for high-dim features whose  order does not affect the mapping output. Regarding the practical aspect, although much practical evidence has shown that DeepSets is indeed hard to learn sometimes, there have been many NN architectures that adopt DeepSets as their key components due to its simplicity and computational friendliness. For example, when GNNs aggregate features from the neighbors of a node, the most commonly used operation follows weighted mapping + sum/average pooling + weighted mapping +... . We believe our analysis also provides some key ingredients to study the functional approximation capability of GNNs, and this topic has a wide range of applications in practice.

---

### Official Review · Reviewer_gdbo · 2023-07-02

**Soundness:** 3 good
**Presentation:** 3 good
**Contribution:** 3 good
**Rating:** 6
**Confidence:** 3

**Summary:**

This manuscript proves that an embedding with polynomial width in the set size and feature dimension is sufficient to precisely reconstruct a set function, under some constraints on the embedding layer architecture. The main contribution is on the upper bound, which removes assumptions of previous studies, and reduces the gap exponentially. The scope of this study covers permutation-invariant and permutation-equivariant set functions in high-dim scenarios.

The key idea of the proof is to construct the embedding vector of the claimed length, and show the injectivity of any input set elements with the proposed embedding layers. The weights are used to (1) save the original features; (2) form an 'anchor' of the set element, which 'shares' the same permutation with the set element features; (3) store the coefficients that mix the original feature and the anchor.

**Strengths:**

- The manuscript is clearly written. The problem is sound and well-motivated.
- The results are significant, extending previous study to high-dimensions and exact representation of sets, in addition to tightening the upper bound from exponential to polynomial.


**Weaknesses:**

The major weakness of this study has been summarized in the limitation section already: (1) There are two important parameters in DeepSet, however only one is considered. (2) The lower bounds are trivial. (3) The upper bound depends on the specific neural network architecture.

Another issue is the proof is an existential argument. That is if we find those weights, the network can precisely represent the set function. However it is not clear if the weights belong to some space as a convergence by training the network. If the authors could provide intuition on this it would be good.

It took me longer than expected to understand the functionality of the three segments of the weights, especially the third one. I would suggest one more paragraph describing the mentality of the construction, together with the Eq 5,6,7.

From the technique side, there is no novel technique involved, and this together with the major weakness would be my personal reasonable potential to reject this paper. But fairly speaking I think the depth is OK, and the results are sufficiently impressive. Thus I recommend accept.

**Questions:**

- Line 223 and 340 indicate the lower bound of $L\geq ND$ for the LLE architecture, it may be better to add to line 63.
- In DeepSets, the $\rho$ and $\phi$ function can be implemented by several neural network layers of choice. Is function $\phi$ narrowed down to either 2 layers (LP) or 3 layers (LLE)? Or I believe at least the "linear" layer can be extended to multiple fully-connected layers. More generally, what is the flexibility in the network design during the implementation?
- This is out of curiosity only, I think it would be more intrigued to study the unconditional lower bound of this problem. Theorem 2.4 seems to address this when $D=1$. What is the barrier of obtaining an $\Omega(f(N,D))$ for general $D$?

**Limitations:**

Mentioned above.

---

> ### Author Rebuttal · Authors · 2023-08-10
>
> We sincerely thank reviewer gdbo for acknowledging the depth and significance of our work. Please see our responses as below:
>
> **1. The proof is an existential argument and the convergence of training is not guaranteed.**
>
> This work focuses on expressive power analysis for DeepSets and the main goal is to show such an architecture can represent every function in some class. Such proofs are often constructive and existential similar to many other works on expressiveness analysis for neural networks [1,2,3], including DeepSets [4]. Typically, the optimization procedure for such an architecture is beyond the scope of the study of expressive power. Also, we want to emphasize that the weights for the construction of LP layer that allow good expressive power are actually dense in the parameter space, which are not one particular set of weights and are amenable to optimization.
>
> [1] Vitaly et al. Lower bounds for approximation by MLP neural networks
>
> [2] Yarotsky. Error bounds for approximations with deep relu networks
>
> [3] Yarotsky. Universal approximations of invariant maps by neural networks.
>
> [4] Zaheer et al. Deep Sets.
>
> **2. One more paragraph describing the mentality of the construction.**
>
> With the correction in our general response, the proof idea for both LP and LE layer has been unified. The high-level idea of our construction consists of two major steps: 1) construct anchor with Lemma 4.4, and 2) couple each feature channel with the anchor through the mixing scheme provided by LP and LE, repsectively. To show injectivity, we first 1) invoke Lemma 4.5 and Lemma 4.8 to induce pair-wise alignment with anchors, and then 2) apply Lemma 4.3 to obtain union alignment. We have prepared an anonymous link to the revision, which includes a paragraph and a figure to demonstrate the contruction idea and the connection to each proof techniques.
>
> **3. The proof technique is not novel.**
>
> Our work stands on the shoulder of giants and the overall proof idea follows DeepSets by constructing continuously invertible sum-pooling. However, this result cannot be achieved without new mathematical establishment. To tackle high-dimensional features, we introduce a novel mathematical device: anchor, which is essential to show the injectivity of the sum-pooling and has not been used in the relevant literature, to our best knowledge. Other reviewers also appreciate the novelty behind. Despite intuitively defined, our Lemma 4.3 shows a significant result that by pairwisely coupling each feature columns with the anchor, one can impose the global alignment, which largely reduces the complexity of the construction. Also Lemma 4.4 reveals that such a useful anchor can be easily constructed via elementwise linear mapping. Besides, we invent two pairwise coupling scheme via a special class of linear combination and monomials, corresponding to LP and LE layer, respectively.
>
> **4. What is the flexibility in the network design during the implementation?**
>
> Akin to DeepSets, both $\phi$ and $\rho$ can be implemented by neural networks using their universal approximation property. We also note that construction established by DeepSets also fixes the initantiation of $\phi$ as a power series functions (cf. Lemma 4 in [1]). Our construction serves as the extension to DeepSets' construction to high-dimensional features.
>
> [1] Zaheer et al. Deep Sets
>
> **5. What is the barrier of obtaining an $\Omega(f(N, D))$ for general $D$?**
>
> Proving the lower bound requires another existential argument to find a function which cannot be represented if $L \le f(N, D)$. In [1], authors restrict $\phi$ and $\rho$ to be analytic functions, and find an analytic function which cannot be approximated by any such $\phi$ and $\rho$ without an exponential width. However, it becomes rather challenging to find a counter-example if we relax the constraint on $\phi$ and $\rho$ to arbitrary continuous functions. In this paper, we make an attempt to derive the lower bound by fixing the architecture of $\phi$.
>
> [1] Zweig et al. Exponential Separations in Symmetric Neural Networks

---

### Official Review · Reviewer_zB8R · 2023-07-05

**Soundness:** 1 poor
**Presentation:** 2 fair
**Contribution:** 2 fair
**Rating:** 6
**Confidence:** 4

**Summary:**

The paper studies the representative properties of DeepSets models, which are networks for length-$N$ sequential modeling that apply identical neural networks to each $D$-dimensional input $x^i$ to obtain $L$-dimensional features, sum up their outputs, and apply an additional neural network to the output. While past results have tightly characterized the setting where $D = 1$, positive results for the $D > 1$ case are less explored. A recent lower bound by Zweig and Bruna shows that $L$ must grow exponentially with $\sqrt{N}$ and $D$ in order to approximate any permutation-invariant function under the condition that the networks use analytic activations.

The primary contribution of the paper Theorem 3.1, which claims that there exists a construction with $L = \mathrm{poly}(N, D)$ that exactly represents any permutation-invariant target as a DeepSets model with $L$-dimensional features.

**Strengths:**

The introduction is well-written and the problem is set up in a clear way. If the result is indeed correct, I think it would be an interesting contribution to the literature on DeepSets approximation properties and the limitations of current lower bounding techniques for DeepSets.

**Weaknesses:**

In it's current form, I am concerned that the result is not correct as written. There are two primary issues, which I would like to see addressed if I am to reconsider my score for the paper.

First, I think the proof of the second part of Theorem 3.1 is false as written. The proof relies on Lemma 4.9, which claims that parirwise alignment is sufficient for union alignment, and as far as I am aware, is not proved in the paper or the appendix. I believe Lemma 4.9 to be false.

Consider the following counter-example. For $D = 3$ and $N = 4$, let $x_1 = (0, 1, 1, 0), x_2 = (0, 0, 1, 1), x_3 = (0, 1, 0, 1) \in \mathbb{R}^N$ and $x_1' = (1, 0, 0, 1), x_2' = (1, 1, 0, 0), x_3' = (1, 0, 1, 0)$. Note that $X = [x_1^T \ x_2^T \ x_3^T ] \not\sim X' = [x_1'^T \ x_2'^T \ x_3'^T]$, since the first row of $X$, $x^1 = (0, 0, 0)$ does not belong to any row of $X'$. However, the two are pairwise aligned.
* Note that $[x_1^T \ x_2^T] \sim [x_1'^T \ x_2'^T]$, with the permutation $\sigma = (1 \ 3) (2 \ 4) $ as witness.
* $[x_1^T \ x_3^T] \sim [x_1'^T \ x_3'^T]$ with permutation $\sigma = (1 \ 2) (3 \ 4)$.
* $[x_2^T \ x_3^T] \sim  [x_2'^T \ x_3'^T]$ with permutation $\sigma = (1 \ 4) (2 \ 3)$.

Since the proof relies on this lemma, the proof of the lemma does not appear, and there exists a simple counter-example, I am unable to accept second part of the main theorem as true.

Second, while I have not pinpointed any major technical flaws with the proof of the first bullet, I am a unsure how the result does not contradict the lower bound on Zweig and Bruna. The theorem claims that the LP architecture (which consists of features with polynomial activations and continuous function that inverts the embedding) with $L = O(N^5 D^2)$ is sufficient to exactly represent any continuous permutation-invariant $f: \mathbb{R}^{N \times D} \to \mathbb{R}$. Theorem 3.4 of ZB suggests that if $L \leq N^{-2} \exp(O(\min(D, \sqrt{N}))$, then there exists some analytic $g$ such that any DeepSets model $f$ with feature dimension $L$ and arbitrary NNs with analytic activations cannot approximate $g$.

As far as I understand both results, the only way these two results do not represent a contradiction would be if the continuous function that inverts the embedding $\rho$ is non-analytic, and hence cannot be represented the networks described by ZB. However, if our input $x$ is from a compact set (like the complex circle, from ZB) (and hence, the aggregated polynomial features $\sum_i \phi(x^i)$ belong to a compact set) and $\rho$ is continuous, then the universal approximation results for two-layer neural nets imply the existence of a sufficiently wide 2-layer neural network with analytic activations (like the sigmoid) that approximates $\rho$ arbitrarily well. I think this would contradict the ZB result on inapproximability.

All that said, it's possible that I've overlooked or misunderstood something. If there is no contradiction or if the authors believe that the result by ZB is incorrect, please let me know, and I'd be happy to discuss further.

**Questions:**

My questions are addressed above.

**Limitations:**

My perceived limitations of the paper are detailed above. Beyond concerns over correctness, the theoretical results are direct about their limitations.

---

> ### Author Rebuttal · Authors · 2023-08-10
>
> We sincerely thank reviewer zB8R for your careful examination of our work and raise meaningful discussion. We have corrected all the flaws in our results and proofs, and all the modifications are summarized in our **general response**. Please read our detailed response as below:
>
> **1. Errors of Lemma 4.9 for LLE architecture.**
>
> After submission, we also noticed this flaw in our proof. We have corrected our proof in the new revision. Technically, instead of adopting Lemma 4.9 for union alignment, we equip our construction with anchor construction, and thus we can first utilize Lemma 4.3 to obtain $[x_i, a] \sim [x_i', a'], \forall i \in [N]$ and leverage Lemma 4.8 to reach $[x_1, \cdots, x_N] \sim [x'_1, \cdots, x'_N]$. We have prepared an anonymous link to the full version of the corrected proof. According to the rebuttal policy, we can post the link upon your request. We would sincerely appreciate it if you could leave a chance to re-assess our hard work.
>
> **2. Contradiction to ZB's result.**
>
> To our best knowledge, the universal approximation theorem with the analytic activation function (e.g., sigmoid) [1] is established over a compact disk $[0, 1]^L$. However, we note that $\rho$ is a continuous function over the domain $\mathcal{Z} = \{\sum_{i} \phi(x^{(i)}): X \in \mathbb{R}^{N \times D}\}$ whose topological structure may differ from a disk. Although $\mathcal{Z}$ is in the ambient space $\mathbb{R}^L$ and $\mathcal{Z}$ is compact if the input $X$ is from a compact space, it does not mean any continuous functions defined over $\mathcal{Z}$ can be universally approximated by NNs with analytic activations. Moreover, the function being continuous over $\mathcal{Z}$ does not mean it has a continuous extension over $[a,b]^L$, but the universal approximation theorem for analytic activations is applied to continuous functions over $[a,b]^L$. An example of missing such continuous extension is as follows: Consider a function defined on rational numbers in $[0,1]$, i.e., $Q_{[0,1]}$, $f(x)= 0$ if $x < 1/\pi$ and $f(x)= 1$ if $x > 1/\pi$. $f(x)$ is continuous in $Q_{[0,1]}$ while $f(x)$ does not have a continuous extension over $[0,1]$. Hence we doubt that universal approximation theorem with analytic functions may not be applicable to $\rho$ defined over $\mathcal{Z}$. We will include this remark into our revision.
>
> [1] Cybenko, G. "Approximation by superpositions of a sigmoidal function.

---

> > ### Comment · Reviewer_zB8R · 2023-08-11
> >
> > > 1. **Errors of Lemma 4.9 for LLE architecture.**
> >
> > I appreciate the authors' detailed response, and I would be happy to review the full version of the corrected proof. Would you mind sending me the anonymous link?
> >
> > > 2. **Contradiction to ZB's result.**
> >
> > I am not an analysis expert, so it's possible that I am wrong about this. (And if any other reviewers have a background in analysis or topology, I'd appreciate their perspective.) I'm still not entirely convinced that the above argument rules out the application of the UAT to $\rho$.
> >
> > > To our best knowledge, the universal approximation theorem with the analytic activation function (e.g., sigmoid) [1] is established over a compact disk.
> >
> > There are several presentations of the UAT, and the presentation by [Hornik, Stinchcombe, and White](https://deeplearning.cs.cmu.edu/F21/document/readings/Hornik_Stinchcombe_White.pdf) applies to general compact sets of $\mathbb{R}^L$; see their Theorem 2.4.
> >
> > >  Moreover, the function being continuous over $\mathcal{Z}$ does not mean it has a continuous extension over $[a, b]^L$.
> >
> > I don't believe that this is true. By the [Tietze extension theorem](https://en.wikipedia.org/wiki/Tietze_extension_theorem), a continuous function mapping a closed subset of a normal space to the real numbers has a continuous extension to the normal space. Because all compact sets are closed and $[a, b]^L$ is normal, a continuous extension exists.
> >
> > For the case you mentioned, $Q_{[0, 1]}$ is not compact because it is not closed. Hence, the non-existence of a continuous extension does not contradict $\mathcal{Z}$ having an extension.

---

> > > ### Comment · Reviewer_S2JM · 2023-08-11
> > >
> > > I don't think there's a contradiction to the ZB result, because it's working with complex analytic functions.  In the complex domain setting, $\rho$ is still (I believe) continuous but not necessarily analytic.  And universal approximation results don't apply for approximating non-analytic functions by analytic NNs, as seen from the classical fact that you can't uniformly approximate 1/z over the complex unit circle by any analytic function.

---

> > > ### Author Response · Authors · 2023-08-12
> > > **Follow-up Response to Reviewer zB8R**
> > >
> > > Dear Reviewer zB8R,
> > >
> > > We are grateful for your rigor which leads to our in-depth and meaningful discussion.
> > >
> > > **1. The full version of the corrected proof.**
> > >
> > > According to the rebuttal policy, we have sent the anonymous link to AC via an official comment only visible to him/her. We would appreciate it if AC could help share the link to all the reviewers. We apologize for the haste when prepareing this manuscripts, and any extra workload caused to you. Following subsequent rigorous efforts, we managed to produce a correct and enhanced version, which still supports our main claim while further relieving some 'strict' assumptions.
> > >
> > > **2. Contradiction to ZB's results.**
> > >
> > > We sincerely appreciate reviewer S2JM's reply, which reminds us all the results in [1] are established for the complex domain, where the existing UAT may not be applicable when limited to analytic activations. In a summary:
> > >
> > > * ZB's work [1] does not specify any results for the real domain. Since it remains intractable to extend their proof technique to real values, our results on real domain should not contradict ZB's results by any means.
> > > * Our result on complex domain still does not contradict ZB's conclusion. This is because UAT with analytic activation is not applicable to complex domain, thus, one cannot approximate the non-analytic $\rho$ with analytic neural networks. Reviewer S2JM has given an example 1/z which cannot approximated by analytic functions, while 1/z is not defined at $z=0$. $f(z) = \exp(-1/z^2)$ with $f(z)=0$ if $z=0$ is another example that is continuous over the entire complex disk but cannot be approximated by analytic functions. Note that complex analytic functions are not dense in the space of continuous functions. Overall, from these examples, we can see that ZB's results are limited to complex analytic functions and may not be applied to the real domain. Moreover, the requirement on using merely analytic activations in the complex domain is a very strong requirement, which on the other hand indicates the significance of our results that analyze the case without this requirement.
> > >
> > > We will include these important remarks into our revision.
> > >
> > > [1] Zweiga et al., Exponential Separations in Symmetric Neural Networks

---

> > > > ### Comment · Reviewer_zB8R · 2023-08-12
> > > >
> > > > Thank you both for the clarification; I appreciate the thoroughness of the above comments. Upon further inspection, I agree that my interpretation of the ZB results was incorrect, due to my misunderstanding of the differences of universal approximation between real and complex-valued functions. As such, I no longer see such a contradiction between the results of the two papers. After seeing the corrections on the anonymous link (should the AC pass them along), I will happily reconsider my score.

---

> > > > > ### Author Response · Authors · 2023-08-18
> > > > >
> > > > > Dear Reviewer zB8R,
> > > > >
> > > > > We thank you again for the rigourous examination of our submission. To help you re-evaluate our work, an anonymous link to our revised results has been posted as an AC message few days ago. We would like to ascertain whether you have had the opportunity to review the amended proofs. If so, we would be more than happy to provide more information or clarification, should it be necessary. If the link is not allowed to be disclosed, we are inquiring about the specific details we can provide to facilitate your comprehensive examination and reassessment of our work. We sincerely wish to bring our theoretical advances on set function expressive power analysis into this community.
> > > > >
> > > > > Best,
> > > > >
> > > > > Paper 3792 Authors

---

> > > > > > ### Comment · Reviewer_zB8R · 2023-08-20
> > > > > >
> > > > > > Given that I have not received the anonymous link to the updated draft, I will tentatively raise my score, due to the results of the previous discussion addressing my primary concerns. Should I receive the manuscript later in the discussion process, this score is subject to change.

---

> > > > > > > ### Author Response · Authors · 2023-08-20
> > > > > > > **Thanks for your generosity!**
> > > > > > >
> > > > > > > Dear Reviewer zB8R,
> > > > > > >
> > > > > > > We are grateful for your generosity, open-mindedness, and extra workload on assessing our work. Acknowledging the difficulty of evaluating the correctness of the linear+exponential (LE) architecture without access to the complete revised manuscript, we would like to emphasize that our presentation of the linear+power mapping (LP) architecture in the current submission remains accurate and significantly enough. This LP architecture effectively reduces the exponential bound to a polynomial one, which substantiates our principal assertion independently of the LE archtecture.
> > > > > > >
> > > > > > > Furthermore, we would like to highlight that our newly introduced LE architecture not only enhances this polynomial bound but also rectifies the proof and eliminates the positivity assumption inherent in the original LLE architecture as presented in the current version. We sincerely hope AC could make its full presentation visible to the reviewers before the end of the rebuttal session.
> > > > > > >
> > > > > > >
> > > > > > > Best,
> > > > > > >
> > > > > > > Paper 3792 Authors

---

### Official Review · Reviewer_YiWd · 2023-07-31

**Soundness:** 3 good
**Presentation:** 2 fair
**Contribution:** 2 fair
**Rating:** 5
**Confidence:** 3

**Summary:**

This paper studies the required neural network width for representing permutation invariant functions on sets. Existing works either focus on the case where the set elements are scalars, or require exponentially large neural network width with respect to the dimensions of the set elements. This work proves that, under certain assumptions, both the upper bound and lower bound on the required neural network are polynomial in the set size and set element dimensions.

**Strengths:**

- The authors show that the bounds in this work are significantly better than existing results.

- This result implies that moderately wide networks are expressive enough to represent set functions.

**Weaknesses:**

- **Lack of discussions on the practical implications of the results.** It seems that the main idea in the proof is to choose the mapping $\phi$ in a clever way. I'm curious about the practical implications of the constructions in the proof. Particularly, the authors mention _"neural networks to learn a set function have found a variety of applications in particle physics, computer vision and population statistics"_. I recommend the authors discuss how the results in this paper can provide insights and improvements on some typical network models in those areas (e.g., GNN, PointNet).

- **Lack of experimental verification.** Although the bounds in this work look significantly better than existing results, experimental verification would strengthen the argument.

- **Presentation issues.** I believe the authors should polish the presentation of Table 1 (comparisions with exsiting results), and the statement of the main theorem (Theorem 3.1)
   - Table 1:
      + $D+1$ and $D$ should be $N+1$ and $N$.
      + Use big-O notation to present the results of Segol et al. and Zweig & Brun.
      + I recommend to present the upper bound and lower bound separately as two columns.
      + The meaning of "Exact Rep." should be explained in the caption.
   - Theorem 3.1:
      + In lines 167 & 169, "For some" should be deleted as these two lines are part of the _where_ clause.
      + Eq. (2) $w_1x\to w_1^{\top}x$. Otherwise please specify that $w_1, ..., w_K$ are row vectors.
      + The main result contains both upper bound and lower bound. I recommend stating these two separately. Particularly, the lower bound should be stated as a negative result. Theorem 2.4 in [1] is a good example of how the result should be rigorously formulated.
      + In the LLE architecture setting, one more assumption $\mathcal{K}\subseteq \mathbb{R}_{>0}$ is required. The authors should highlight this as a premise of the theorem, instead of putting it in the _where_ clause.

      [1] Aaron Zweig and Joan Bruna. Exponential separations in symmetric neural networks.
- **Minor errors**.
   - Line 28: DeepSets$\ \ $[9] $\to$ DeepSets [9] (There are two spaces between "DeepSets" and "[9]" in the submission).
   - Line 117: suggests $\to$ suggest.

**Questions:**

- Can you discuss how the results in this paper can provide insights and improvements on some typical network models in those areas (e.g., GNN, PointNet)?

- Can you provide experimental verification of the result?

- Lines 189-191 state that the assumption $\mathcal{K}\subseteq \mathbb{R}_{>0}$ is not essential. Then why not remove the assumption and present a stronger result?

**Limitations:**

The limitations have been discussed in lines 375-377.

---

> ### Author Rebuttal · Authors · 2023-08-10
>
> We thank reviewer YiWd for acknowledging the significance of our results compared with prior arts. We will carefully proofread our paper and fix all the typos. Per your questions, please read our responses below:
>
> **1. Practical implication of the results (e.g., in GNN, PointNet).**
>
> The LP layer extends the construction of the original DeepSets by prepending a linear layer before its power series activation (Lemma 4 in [1]). In addition, we propose the LE layer (a improved version of LLE, see our general response), which leverages more commonly used components   in deep learning, such as linear and exponential layers.
>
> Both GNNs and PointNet require set representation in their architecture design. GNNs learn to pass information along graph topology via a neighorhood aggregation operation at each layer of computation, which essentially corresponds to a set function [2]. PointNet processes point clouds by representing points as an unordered set and follows the DeepSets-like architecture. Our work gives a rigorous justification that moderately many neurons are sufficient to represent a high-dimension set with DeepSets. This affirms the feasibility of DeepSets architecture given high-dimensional features and explains why DeepSet-like operations can be effectively adopted in GNN and PointNet.
>
> [1] Zaheer et al., Deep Sets
>
> [2] Xu et al., How Powerful are Graph Neural Networks?
>
>
> **2. Lack of experimental verification.**
>
> To verify our theoretical claim, we conducted proof-of-concept experiments. Similar to [1], we train a DeepSets with $\phi$ and $\rho$ parameterized by fully connected neural networks to fit a function which takes the median over a vector-valued set according to the lexicographical order. The input features are sampled from a uniform distribution. The critical width $L$ is taken at the point where RMSE first reaches below 10% above minimum value for this set size. The relationship between $L$ and $N, D$ is plotted in the attached PDF. We observe $\log(L)$ grows linearly with $\log(N)$ and $\log(D)$, which validates our theoretical claim.
>
> [1] Wagstaff et al., On the Limitations of Representing Functions on Sets
>
> **3. Why not remove the assumption (Lines 189-191) and present a stronger result?**
>
> Thanks for this suggestion. After submission, we have been working on improving our results. The revised results are presented in our general response. We replace LLE layer with linear+exponential (LE) layer, which no longer requires the assumption on positivity feature space.

---

> > ### Comment · Reviewer_YiWd · 2023-08-12
> > **Thanks for the rebuttal**
> >
> > I appreciate the authors' clarification on the practical implications and additional experiments. Regarding the theorem, it seems that the updated result still mixes the upper bound and low bound in one statement, and the whole theorem is a lengthy sentence with the key result appearing in the clause (correct me if I'm wrong). I'd recommend the authors to try to further improve the clarity. Besides, I will consider change my rating if the correctness of the new result can be checked.

---

> > > ### Author Response · Authors · 2023-08-16
> > > **Thanks for the reply**
> > >
> > > Dear Reviewer YiWd,
> > >
> > > We appreciate you prompt reply and a reminder on the presentation form. We have sent an external link to our new results to AC. We will also take your advice to present our main results in our final version which provides better clarity. Thanks!
> > >
> > > Best,
> > >
> > > Paper 3792 Authors

---

> > > ### Author Response · Authors · 2023-08-20
> > >
> > > Dear Reviewer YiWd,
> > >
> > > We want to thank you again for the constructive comments on improving the presentation of this paper. To help you finalize your rating as the rebuttal period is ending soon, we provide the revised format of our main result following your instructions as below, which divides the upper and lower bounds into two statements. We are more than glad to hear more suggestions from you to further enhance the clarity.
> > >
> > > **[The main result]** Suppose $D\geq 2$ and given any continuous permutation-invariant function $f: \mathbb{R}^{N \times D} \rightarrow \mathbb{R}$. Consider a **continuous** mapping $\phi: \mathbb{R}^{D} \rightarrow \mathbb{R}^{L}$ such that either of the following holds:
> > > * For some $L \le N^5D^2$, $\phi$ admits *linear layer + power mapping (LP)* architecture:
> > > $$
> > > \phi(x) = \begin{bmatrix}
> > > \psi_N(w_1^\top x)^{\top} & \cdots & \psi_N(w_{K}^\top x)^{\top}
> > > \end{bmatrix}^\top
> > > $$
> > > for some $w_1, \cdots, w_{K} \in \mathbb{R}^{D}$, and $K = L / N$.
> > > * For some $L \le N^4D^2$, $\phi$ admits *linear layer + exponential activation (LE)* architecture:
> > > $$
> > > \phi(x) = \begin{bmatrix}
> > > \exp(v_1^\top x) & \cdots & \exp(v_L^\top x)
> > > \end{bmatrix}^\top
> > > $$
> > > for some $v_1, \cdots, v_L \in \mathbb{R}^{D}$.
> > >
> > > Let $\mathcal{Z} = \{ \sum_{i} \phi(x^{(i)}) : X \in \mathbb{R}^{N \times D}\} \subseteq \mathbb{R}^L$ be the range of $\phi$. Then there exists a **continuous** mapping $\rho: \mathcal{Z} \rightarrow \mathbb{R}$, such that for every $X \in \mathbb{R}^{N \times D}$, $f(X) = \rho\left(\sum_{i=1}^{N} \phi(x^{(i)}) \right)$.
> > >
> > > Moreover, if $\phi$ admits *linear layer + power mapping (LP)* architecture while $L < N(D+1)$, $f(X) \ne \rho\left(\sum_{i=1}^{N} \phi(x^{(i)}) \right)$ for any continuous $\rho: \mathcal{Z} \rightarrow \mathbb{R}$.
> > >
> > > To help check the correctness, we had already sent our amended results and proofs to AC few days back. However, as these materials have not yet been shared with the reviewers (as indicated by reviewer zB8R), we are reaching out to inquire if there are any additional pertinent details we can furnish to facilitate your assessment.
> > >
> > > Furthermore, we wish to highlight that the proof of our LP architecture presented in the current submission is inerrant. This underscores that our primary assertion remains valid, even if, in a worst-case scenario, the examination of the correctness of the LE architecture cannot be finished during the rebuttal phase.
> > >
> > > We would greatly value any additional endeavors that might contribute to a fair and hopefully more positive assessment to our work. Many thanks in advance.
> > >
> > > Best,
> > >
> > > Paper 3792 Authors

---

### Author Rebuttal · Authors · 2023-08-10

We sincerely appreciate all the reviewers for their time and efforts reviewing our paper. However, we have to apologize for the misplacements in the current submission due to a tight timeline when we prepared this manuscript, which led to an imprecise statements. After we submitted our paper, we noted the incorrectness of Lemma 4.9 (also revealed by Reviewer zB8R), so the statement on LLE in Theorem 3.1 is incorrect in its current form. Afterwards, we have been working on fixing the proof and successfully fixed the relevant error behind in July. Consequently, a corrected result is presented as below. **Our main claim polynomially many neurons are sufficient still holds for both architectures investigated.** We also improve this result by relaxing the assumptions.

**[The main result]** Suppose $D\geq 2$. For any continuous permutation-invariant function $f: \mathbb{R}^{N \times D} \rightarrow \mathbb{R}$, there exist two continuous mappings $\phi: \mathbb{R}^{D} \rightarrow \mathbb{R}^{L}$ and $\rho: \mathcal{Z} \rightarrow \mathbb{R}$, where $\mathcal{Z} = \{ \sum_{i} \phi(x^{(i)}) : X \in \mathbb{R}^{N \times D}\}$, such that for every $X \in \mathbb{R}^{N \times D}$, $f(X) = \rho\left(\sum_{i=1}^{N} \phi(x^{(i)}) \right)$ where
* For some $L\in [N(D+1),N^5D^2]$ when $\phi$ admits *linear layer + power mapping (LP)* architecture:
$$
\phi(x) = \begin{bmatrix}
\psi_N(w_1^\top x)^{\top} & \cdots & \psi_N(w_{K}^\top x)^{\top}
\end{bmatrix}^\top
$$
for some $w_1, \cdots, w_{K} \in \mathbb{R}^{D}$, and $K = L / N$.
* For some $L\in [ND,N^4D^2]$ when $\phi$ admits *linear layer + exponential activation (LE)* architecture:
$$
\phi(x) = \begin{bmatrix}
\exp(v_1^\top x) & \cdots & \exp(v_L^\top x)
\end{bmatrix}^\top
$$
for some $v_1, \cdots, v_L \in \mathbb{R}^{D}$.

We enumerate the differences with the previous results, and a few other remarks as follows:

1. The bounds for LP architectures are correct and kept unchanged.
2. For the LLE layer, we are able to simplify it to a linear + exponential (LE) layer. Essentially, the LE layer first transforms feature space via an exponential function, and then compounds anchor construction (Lemma 4.4) with LLE (see 4). **With this modification, the new result removes the positive input constraint.**
3. We adjust the upper bound for LE to $N^4D^2$. **Such a bound is still polynomial in feature dimension and set length, and tighter than the LP layer.**
4. The proof technique becomes unified for both LP and LE layer. Dismissing the reliance on Lemma 4.9, we derive the current result by combining Lemma 4.8 with the anchor alignment argument (Lemma 4.3). Specifically, our construction and proof outlines are listed as below:
    1. We first cast an LE layer into linear + exponential + LLE layer: $\exp(V^\top x)=\exp(U^\top \log \exp(\Omega^\top x))$ where $V = \Omega U$. Note that here $\exp$ and $\log$ are entry-wise operations.
    2. Let $\Omega = [W^{(1)}, W^{(2)}] \in \mathbb{R}^{D \times (D+K_1)}$, $K_1 = N(N-1)(D-1)/2+1$ follow the anchor construction as in Sec. 4.1.
    3. Let $U = [\cdots, u_{i,j,p,q}, \cdots] \in \mathbb{R}^{(D+K_1) \times L}$, where $u_{i,j,p,q} = (q-1) e_i + (p - q + 1) e_{D+j}$, $i \in [D], j \in [K_1], p \in [N], q \in [p+1]$. Thus, we have $L=DK_1N(N+3)/2 \le N^4D^2$. With such construction, we can enumerates all bivariate monomials between feature channels and anchor with degree less or equal to $N$: $\exp(u_{i,j,p,q}^{\top} \log(x)) = x_i^{q-1} a_j^{p-q+1}$.
    4. Injectivity can be shown by: 1) Lemma 4.8 indicates that $\sum_i \phi(x^{(i)}) = \sum_i \phi(x'^{(i)}) \Rightarrow [\exp(x_i), \exp(a_j)] \sim [\exp(x'_i), \exp(a'_j)], \forall i \in [D], j \in [K_1]$, 2) Lemma 4.3 induces $[\exp(x_i), \exp(a_j)] \sim [\exp(x'_i), \exp(a'_j)], \forall i \in [D], j \in [K_1] \Rightarrow \exp(X) \sim \exp(X') \Rightarrow X \sim X'$, where we note $\exp$ preserves the properties of anchor and permutation equivalence class.
    5. Continuity is concluded by: 1) using the same argument in Sec. 4.3 to establish continuous inverse from $\sum_i \phi(x^{(i)})$ to $\exp(X)$, and then 2) composing a logarithm to inverse the exponential function.


Due to the page limit, we regret that we could not include all the details in the rebuttal. However, to address this, we have diligently prepared a comprehensive revision with refined text and detailed proofs. An anonymous link is readily available for sharing these additional materials. We would sincerely appreciate it if reviewers are willing to spend time examining and re-evaluating our hard work with the corrected results.

---

### Author Response · Authors · 2023-08-21
**Anonymous Link to Corrected Results and Proofs**

Dear Reviewers and ACs,

We extend our apologies once more for the proof-related artifacts present in our current submission. In response to these errors, we have dedicated significant effort to rectify the issues existing in our current version. The revised results and proofs not only validate our main claims but also eliminate the assumptions inherent in the initial version.

While we previously submitted our revised manuscript to the AC, it appears that these revisions have not yet been made accessible to the reviewers. In order to ensure that the reviewers are able to thoroughly examine our results, we have to provide an anonymous link below. This link directs to a streamlined version of our work, exclusively containing the technical statements and proofs:

https://drive.google.com/file/d/1_rycsT3tSqudJ2VetMo3-LIREJYmRE9Y/view?usp=sharing

We once again express our gratitude for the valuable discussions and interactions during the rebuttal session, as well as for the diligent efforts invested throughout the entire reviewing process. We sincerely hope to contribute our advancements in set representation power analysis to the NeurIPS community.

Best regards,

Paper 3792 Authors

---

### Decision · Program_Chairs · 2023-09-21

**Decision:**

Reject

**Comment:**

This paper analyzes neural network models for representing permutation-invariant functions on sets, and argues that polynomial width is sufficient with high-dimensional features, improving on the exponential bounds of prior works.

The reviewers found the paper to be clear, well-motivated, and relevant for the NeurIPS community. The strengths of the paper include significantly better bounds than existing results, which would imply that even reasonably wide networks are capable modeling set functions. Some weaknesses identified in the discussion include a lack of experimental verification and insufficient discussion of the implications of the results or their practical algorithmic attainability. In light of these considerations alone, the paper is borderline.

There is another significant weakness identified by one reviewer -- the proof of the main result is incorrect. While the authors claim to have fixed the issue, and that the corrected proof carries through without weakening the theorem, such a major correction requires additional reviewing to verify the correctness of the new material. To their credit, the authors tried to make this additional reviewing possible by supplying a revised draft during the rebuttal period. Unfortunately, the CFP states that "Authors may not submit revisions of their paper or supplemental material," and in fairness to other authors who chose to heed this guidance, it is not appropriate for us to consider the revised content in this decision.

All together, this is a borderline paper with a potentially fatal flaw in the main theorem. While it seems likely this flaw has been fixed, the revised draft should be subject to a thorough reviewing process, which unfortunately will have to be at another venue. I encourage the authors to take the opportunity to also address some of the other reviewer's comments when preparing the new manuscript.